# Impact of domestic and foreign investors on ESG disclosure quality in Chinese listed firms: Divergence or convergence?

Wei Yin [1,2*], Yang Su[1], Berna Kirkulak-Uludag[3], An Yan[4], Ruizhe Wang[1]

**1** School of Economics and Management, Southeast University, Nanjing, China, **2** Fintech and Big Data Laboratory of Southeast University, Nanjing, China, **3** Faculty of Business, Dokuz Eylul University, Izmir, Turkey, **4** Gabelli School of Business, Fordham University, United States of America

* yinwei_seu@seu.edu.cn, yinwei_seu@126.com

## Abstract

Using the data of Chinese listed firms, this study constructs a two-layer complex network of Qualified Foreign Institutional Investors (QFII) and Domestic Institutional Investors (DII) to examine their influence on the quality of corporate ESG disclosures. By analyzing the credibility, consistency, and completeness of the network, it finds that improving a firm's position within the QFII-DII complex network significantly enhances ESG disclosure quality. Key drivers include the knowledge transfer effect, information exchange effect, and market supervision effect. Additionally, executive green awareness and strong internal governance amplify the positive impact of network position on ESG quality. The study also reveals variations in the influence of QFII and DII over time and across industries, with similar overall effects. However, companies with higher centrality in the DII network layer are more likely to engage in ESG greenwashing behavior.

## Abstract

Using the data of Chinese listed firms, this study constructs a two-layer complex network of Qualified Foreign Institutional Investors (QFII) and Domestic Institutional Investors (DII) to examine their influence on the quality of corporate ESG disclosures. By analyzing the credibility, consistency, and completeness of the network, it finds that improving a firm's position within the QFII-DII complex network significantly enhances ESG disclosure quality. Key drivers include the knowledge transfer effect, information exchange effect, and market supervision effect. Additionally, executive green awareness and strong internal governance amplify the positive impact of network position on ESG quality. The study also reveals variations in the influence of QFII and DII over time and across industries, with similar overall effects. However, companies with higher centrality in the DII network layer are more likely to engage in ESG greenwashing behavior.

**Data availability statement:** Relevant data are included in the paper and its Supporting Information files. Quarterly data on institutional investor shareholding and company financial variables are obtained from the China Stock Market & Accounting Research (CSMAR) database, while ESG-related data primarily come from Wind, Huazheng, FTSE Russell, SynTao Green Finance, Bloomberg, and annual reports of listed companies. National-level data are sourced from the World Bank's ESG database.

**Funding:** This work was supported by National Social Science of China Program (24BJY093). the laboratory of Philosophy and Social Sciences at Universities in Jiangsu Province-Fintech and Big Data Laboratory of Southeast University (7514009282). The funders had no role in study design, data collection and analysis, decision to publish, or preparation of the manuscript.

**Competing interests:** The authors have declared that no competing interests exist.

## 1. Introduction

In recent years, companies have increasingly focused not only on their economic performance but also on their Environmental, Social, and Governance (ESG) practices. Strong ESG performance not only enhances a company's financial outcomes and market capitalization but also contributes to its competitiveness and resilience to risks [1–4]. As the importance of ESG factors grows, scholars have explored various aspects that influence corporate ESG performance. Among the key drivers, institutional investors have been identified as a significant force in promoting ESG improvements within companies [5].

Institutional investors, by virtue of their substantial role in the capital markets, encourage firms to prioritize ESG issues through their influence and emphasis on responsible governance. With the deepening of China's capital market reforms, particularly with the Qualified Foreign Institutional Investors (QFII) system, foreign investors have become a crucial part of the Chinese financial market. The State Administration of Foreign Exchange's decision to lift QFII investment quota restrictions in 2019 further enhanced foreign investment participation in China, promoting a more open financial market [6–7]. QFIIs tend to bring more professional governance experience and place greater importance on corporate transparency and social responsibility. This contrasts with Domestic Institutional Investors (DII), who may have different perspectives or priorities regarding corporate governance and ESG matters.

However, the impact of QFII and DII on ESG disclosure quality remains an open question. This study aims to explore the effects of these two investor types, focusing on how their combined behavior within the investment network affects the quality of ESG disclosures by Chinese listed firms. Existing research predominantly centers on ESG ratings, often neglecting the vital process of ESG information disclosure, which is crucial for assessing a company's ESG performance accurately.

The interconnection of institutional investors through shared ownership in various firms fosters networks that can reduce information asymmetry, alleviate financing constraints, and promote cooperation among companies [8–9]. These networks not only enhance corporate governance but also support companies in improving their ESG performance through increased transparency and socially responsible investment practices [10–11]. In China, the rising importance of ESG disclosures has been accompanied by initiatives like carbon neutrality, contributing to a surge in ESG reporting. In 2022, over 1,700 listed companies published ESG reports, representing 34% of all listed firms, with a clear shift from focusing on the quantity of disclosures to the quality of information provided [12]. High-quality ESG disclosures, marked by completeness, accuracy, consistency, timeliness, and credibility, are critical for fostering trust among stakeholders and ensuring effective decision-making.

This study explores how the collective behavior of QFII and DII within the investment network impacts the quality of ESG disclosures among Chinese listed firms. It contributes to the ESG literature by shifting the focus from conventional ESG ratings to the underlying disclosure process. Furthermore, the study adopts a novel methodological perspective by employing social network analysis to investigate the role of institutional investors—specifically QFII and DII—in shaping ESG practices. This network-based

approach offers new insights into mechanisms of knowledge diffusion, information exchange, and market oversight, thereby deepening our understanding of how institutional investors can drive improvements in ESG disclosure quality.

## 2. Literature review

As corporate ESG information disclosure plays a pivotal role for stakeholders, including investors and regulators, understanding the factors that influence ESG disclosures has been a focal point for scholars. At the macro level, the external environment, including natural and institutional factors, significantly impacts the level of ESG disclosure. For example, Huang et al. (2022) [13] found that U.S. companies located near natural disasters responded by improving their ESG disclosures, reflecting strategic adjustments in response to changing investor risk perceptions. Baldini et al. (2018) [14] examined multinational firms and identified a negative correlation between strong political institutions and the level of ESG disclosure, with companies in countries with weak political structures demonstrating higher levels of ESG transparency.

At the meso-market level, market competition has been shown to affect ESG disclosure behavior. Ryou et al. (2022) [15] analyzed the impact of reduced import tariffs, finding that heightened competition increases the proprietary costs of ESG disclosure, leading firms to reduce the frequency and depth of independent ESG reports. At the micro-enterprise level, factors such as firm size, ownership structure, and board characteristics are critical in shaping the extent of ESG disclosure.

Foreign institutional investors (QFII) have been identified as significant players in enhancing ESG performance in China, with non-state-owned enterprises and high-tech firms being particularly responsive to QFII investment [16]. A study conducted by Yoo and Chang (2024) [17] indicates that foreign institutional investors strengthen the ESG performance of Chinese companies, whereas Domestic Institutional Investors (DII) have not shown a significant impact on ESG practices.

Despite the growing use of ESG ratings by investors and academics, significant divergences in ESG ratings among agencies raise concerns. Chatterji et al. (2016) [18] observed large discrepancies in ESG ratings, urging caution among stakeholders relying on these ratings for decision-making. Research has shown that such discrepancies create uncertainty in ESG information, which can negatively affect market efficiency and stability. Wang et al. (2024) [12] analyzed the impact of ESG rating divergence on stock returns in China's A-share market, while Avramov et al. (2022) [19] explored its effects on asset pricing in the U.S. markets, finding that greater divergence leads to higher perceived market risk and reduced investor demand.

While much of the literature focuses on the factors influencing the level of ESG disclosure, fewer studies have examined the quality of ESG disclosures. As ESG rating divergence can have adverse effects on corporate performance and market stability, it is crucial to identify factors that enhance the quality of ESG disclosures. This study seeks to fill this gap by exploring the impact of institutional investors on the quality of ESG information disclosure, further advancing research in the ESG domain.

## 3. Theories and hypotheses

### 3.1. Investor choice preference analysis

This study employs the Random Forest algorithm to explore the investment preferences of Qualified Foreign Institutional Investors (QFII) and Domestic Institutional Investors (DII). The dataset utilized for this analysis is described in Section 4.1, with variable definitions and summary statistics provided in Tables 1 and 3. The methodology is outlined in the following steps:

**Step 1** Data processing and preparation

The first step involves gathering firm-level data pertinent to QFII and DII investment preferences. This dataset includes various firm characteristics, such as financial performance indicators, liquidity, profitability, and governance structure. To ensure the integrity and consistency of the data, data cleaning procedures are conducted using the pandas library in Python. This includes handling missing values, removing outliers, and normalizing the data to prepare it for further analysis.

**Step 2** Random Forest model construction

Following data preprocessing, a random subset of the dataset is selected for training. Using the Bootstrap resampling technique, multiple decision trees are constructed to form the Random Forest model. To enhance model stability and ensure robustness, 1000 decision trees are generated, with features randomly selected for evaluation at each split. This approach helps mitigate overfitting and ensures that the model can generalize well to new data.

**Step 3** Feature importance assessment

Upon completing the model training, feature importance scores are derived using the "feature importance" attribute within the Random Forest Regressor. These scores quantify the contribution of each feature to the model's predictive accuracy by measuring the reduction in "impurity" associated with each feature at every split in the decision trees. In this study, the feature importance scores are utilized to evaluate the relative influence of various firm-level factors (e.g., financial indicators, governance structure) on the investment preferences of QFII and DII.

**Step 4** Result analysis and conclusion

From the results of the random forest analysis, although different types of institutional investors differ in their ranking of the importance of various financial indicators (A detailed description of the relevant indicators is shown in Table 1), both for QFIIs and domestic institutional investors (Fig 1), the 'book value', 'equity checks and balances' and 'return on assets 'are all identified as important characteristics for institutional investors' shareholding. On the other hand, 'ESG score' and 'whether the company is a state-owned enterprise' show lower influence, which reveals that both domestic and foreign institutional investors have their own considerations in their investment choices, but they also have the core indicators that they generally pay attention to.

Book value is often used as a measure of a company's intrinsic value, with a higher book value indicating greater financial stability and long-term profitability, making it attractive to institutional investors. Similarly, a balanced equity structure aligns management decisions with shareholders' interests, enhancing efficiency, transparency, and risk mitigation. ROA, which measures profit generation from assets, also plays a crucial role in investment decisions, as higher ROA signals better operational efficiency. These financial metrics lead institutional investors to favor companies with similar performance, creating an equity network that fosters interconnectedness, information exchange, and potential synergies. This study aims to explore how these investor networks influence the quality of ESG disclosures.

With the rise in ESG disclosure, providing ESG information voluntarily has become a strategy for companies to attract investors. However, due to the absence of a unified disclosure framework and effective regulatory oversight, companies may engage in "greenwashing," where they overstate their ESG commitments in their disclosures to meet market trends.

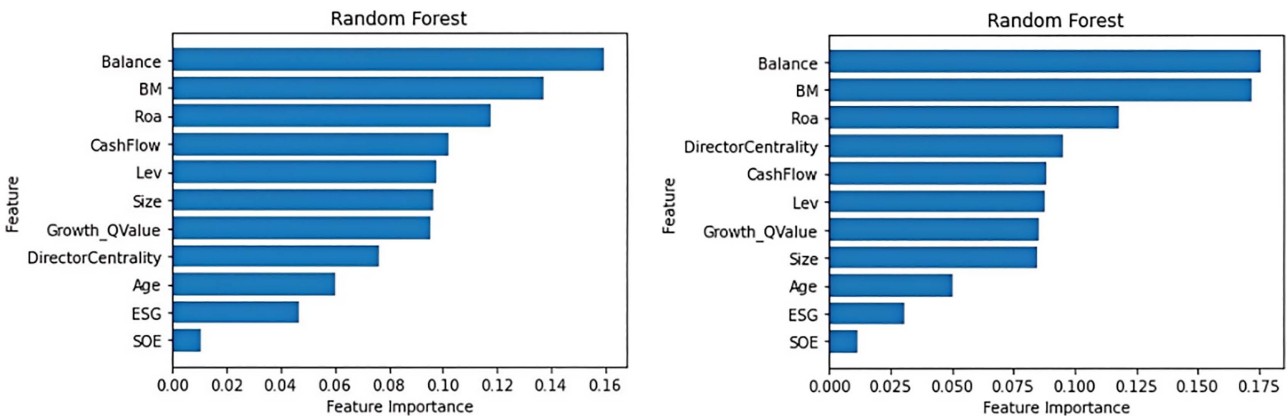

**Fig 1. Feature importance ranking graph based on random forest for QFII (Left) and DII (right).**

**Table 1. A Description of the Features Included in the Feature Importance Ranking.**

| Variable names | Variable symbols | Variable definitions |
|---|---|---|
| Return on Assets | *Roa* | Net profit/Total assets |
| Debt-to-Asset Ratio | *Lev* | Total debts/ Total assets |
| Firm Age | *Age* | Years of business operation |
| Cash Flow Level | *CashFlow* | Net operating cash flows/Total assets |
| Book-to-Market Ratio | *BM* | Net assets/ market capitalization |
| Firm Growth | *Growth_QValue* | Tobin's Q: Market capitalization/Total assets |
| The Nature of Property Rights | *SOE* | Non-state-owned enterprises = 0, state-owned enterprises = 1 |
| Firm Size | *Size* | Natural logarithm of total assets |
| Equity Balance | *Balance* | Proportion of shares held by the second to fifth largest shareholders/ Shareholding of the largest shareholder |
| Board Network Centrality | *DirectorCentrality* | Degree Centrality |
| A company's ESG score | *ESG* | According to the China Securities ESG Rating Index, the ESG performance of enterprises is assigned a value of 1~9 |

This table reports an explanation of the definitions and calculation formulas for the features included in the feature importance rankings.

This discrepancy arises when companies' environmental commitments and public statements exceed their actual practices, leading to a credibility gap in ESG disclosures [20]. Additionally, due to the lack of standardization and transparency in ESG disclosure, various rating agencies may assess the same company differently, leading to inconsistencies in ESG rating results [21]. Furthermore, since the China Securities Regulatory Commission (CSRC) hasn't mandated ESG disclosure for listed companies, except those in specific industries or sectors, the ESG information found in corporate social responsibility or ESG reports is largely qualitative with insufficient quantitative data, resulting in incomplete ESG disclosure.

On the other hand, the shareholder network formed by common institutional investors, namely Qualified Foreign Institutional Investors (QFII) and Domestic Institutional Investors (DII), can drive significant improvements in ESG disclosure quality through its strong network effects. As illustrated in Fig 2, the QFII-DII complex network can influence corporate ESG disclosure quality through three primary channels: the first is knowledge transfer, where the QFII-DII complex network facilitates the communication of ESG expertise and best practices among different members, encouraging a stronger knowledge-sharing effect. The second is complementary information exchange, where the network supports the sharing of domestic and international ESG-related information between QFII and DII, fostering a complementary effect among various institutional investors. The third is market oversight, where the QFII-DII complex network can lead to stronger supervisory effects and improve compliance with ESG standards. The subsequent sections of this paper will delve deeper into these three mechanisms, examining their influence on corporate ESG disclosure quality and discussing strategic measures for enhancing business sustainability.

### 3.2. Hypotheses

From the perspective of knowledge transfer, stakeholder theory posits that stakeholders play a crucial role in the realization of enterprise goals, influencing the survival and development of companies. According to Kallio & Bergenholtz (2011) [22], equity linkages enhance communication and interaction among firms, providing them with opportunities to acquire advanced knowledge and innovative technologies through connections with external organizations. As firms gain prominence within the network, their connections diversify, granting them access to a wider range of knowledge resources. This centrality within the network enables firms to acquire advanced technologies and cutting-edge knowledge. In this context, the entry of QFII brings a global perspective and expertise in ESG, enabling domestic companies to adopt international

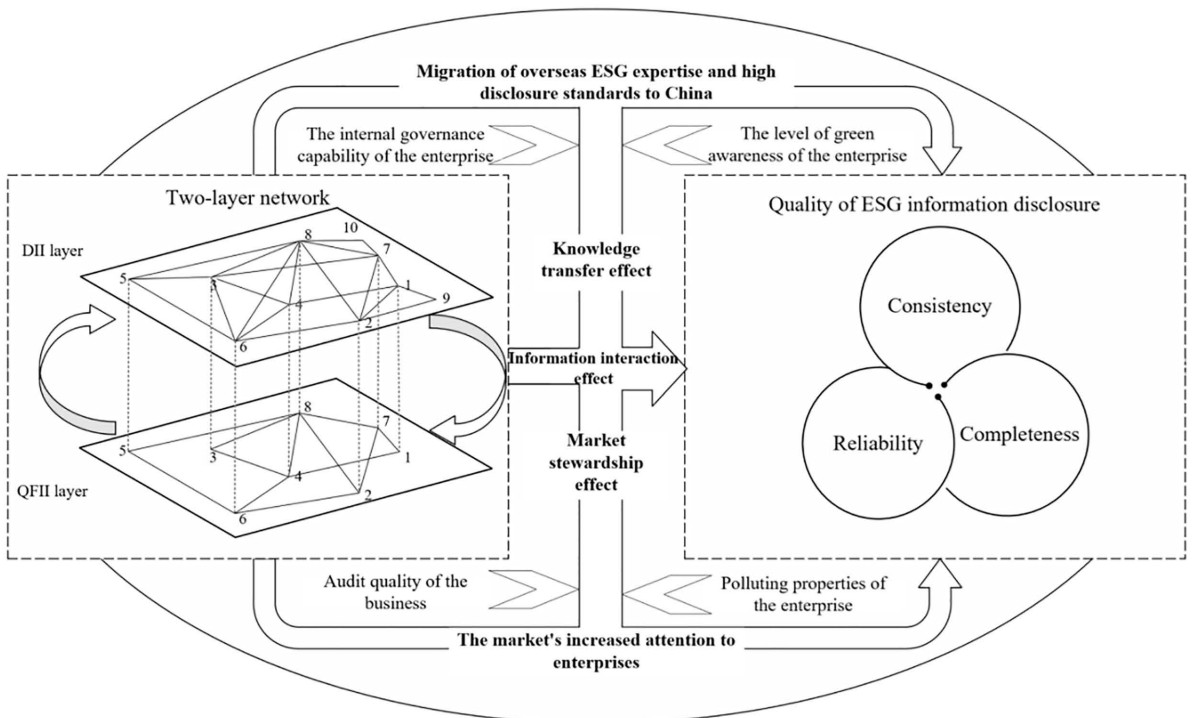

**Fig 2. The Mechanism and Heterogeneity of the Impact of the QFII-DII Complex Network on Corporate ESG Information Disclosure Quality.**

best practices in ESG management and disclosure. This knowledge transfer, originating from QFII's experience, can influence other firms within the network, fostering improved ESG information disclosure standards among domestic companies. Thus, based on knowledge transfer effect, the following hypothesis is proposed:

**Hypothesis 1 (H1):** The QFII-DII complex network enhances the quality of ESG information disclosure among companies by facilitating knowledge transfer, particularly for firms situated at the network's core.

Within the QFII-DII complex network, information exchange plays a pivotal role in improving ESG disclosure practices. Information exchange theory emphasizes the reciprocal flow of information between different stakeholders. In the context of QFII and DII, this bidirectional exchange allows both parties to share insights into international and local ESG practices. While QFII brings knowledge of global ESG standards, DII offer insights into domestic market dynamics, regulatory environments, and cultural nuances. According to Kim et al. (2020) [23], foreign institutional investors, such as QFII, often face information asymmetry in the host country's market. The inter-layer connections within the QFII-DII network facilitate the exchange of market knowledge, allowing QFII to better understand local conditions and adapt international ESG standards accordingly. At the same time, QFII imparts their experience with ESG practices to DII, fostering a mutual understanding that drives both parties to improve ESG disclosure quality. Based on information exchange effect, the following hypothesis is proposed:

**Hypothesis 2 (H2):** The QFII-DII complex network fosters a knowledge exchange effect, wherein companies at the network's core gain access to a rich flow of information that contributes to higher quality ESG information disclosure.

Signaling theory suggests that institutional investors, due to their professionalism and influence in capital markets, can increase market scrutiny on the companies they invest in [24]. This scrutiny often translates into heightened attention from

market participants and analysts, leading to greater pressure on companies to maintain high standards of governance and transparency [25]. Companies with central positions in the QFII-DII complex network, which is characterized by shared institutional ownership, tend to attract more media and market attention. As this external focus intensifies, companies become more conscious of their public image and brand reputation, recognizing the need for high-quality ESG information disclosure. According to signaling theory, robust ESG disclosures not only enhance a company's reputation but also attract more stakeholders, including consumers, employees, and investors, thus reinforcing the company's market position. Therefore, the centrality of a company within the QFII-DII network, accompanied by increased market attention, serves as an incentive for improved ESG disclosure. Building on market oversight effect, the following hypothesis is formulated:

**Hypothesis 3 (H3):** The QFII-DII complex network creates a market oversight effect, where companies at the network's core position experience greater market attention and, as a result, disclose higher-quality ESG information.

The bidirectional information flow between QFII and DII within the complex network not only enhances mutual understanding but also encourages the convergence of ESG standards. As QFII shares international ESG knowledge with DII and DII provides insights into local market conditions, both parties collaboratively work towards improved ESG disclosure standards [26]. This convergence effect is reinforced by the shared ownership and interaction between QFII and DII, leading to greater alignment in ESG practices. This enhanced cooperation fosters a shared understanding of ESG expectations and drives improvements in ESG information disclosure across the network. Hence, based on information flow and ESG standard convergence, the following hypothesis is proposed:

**Hypothesis 4 (H4):** The QFII-DII complex network promotes the convergence of ESG standards through enhanced information exchange, leading to collaborative improvements in ESG information disclosure quality.

## 4. Methodology and data

### 4.1. Sample selection and data sources

Considering QFIIs didn't officially entered China until 2003, this study selects Shanghai and Shenzhen A-share listed companies from 2013 to 2022 as the research sample, in order to fully reflect the role of QFII while incorporating a wide range of institutions and companies. After excluding companies labeled as ST and *ST and those with significant missing data in key variables, the final dataset comprises 37,260 firm-year observations. Quarterly data on institutional investor shareholding and company financial variables are obtained from the China Stock Market & Accounting Research (CSMAR) database, while ESG-related data primarily come from Wind, Huazheng, FTSE Russell, SynTao Green Finance, Bloomberg, and annual reports of listed companies. National-level data are sourced from the World Bank's ESG database. Data computation, analysis, and visualization are conducted using Gephi, Stata 17, and Origin.

### 4.2. The construction of complex networks

Current studies typically measure QFII and DII behavior through ownership structures, trading volume, and institutional investment patterns. According to the investment behavior is a complex and interact behaviors, these measures may not fully capture the complexities of QFII and DII behavior in a networked environment. This paper considers different types of common institutional investor shareholding as a multi-layered network system and uses complex network theory to represent the network of Qualified Foreign Institutional Investors (QFII) and Domestic Institutional Investors (DII), which try to capture not only the direct financial connections between investors and firms but also the cross-layer interactions that reflect shared interests, investment strategies, and influence across domestic and foreign markets.

Referencing the study by Crane et al. (2019) [27], this paper defines that if any two companies $i$ and $j$ have the same domestic institutional investors with at least a 5% shareholding (QFII does not require a 5% threshold), then a network

connection is established between company *i* and *j,* and ultimately creates a multi-layered network with companies as nodes and common institutional investor shareholding relationships as edges. The mathematical model is described as follows:

$$G^{double} = G^U + G^L$$

$$V^{double} = V^U + V^L$$

$$E^{double} = E^U + E^C + E^L$$

$$W^{double} = W^U + W^C + W^L \tag{1}$$

A weighted undirected common institutional investor complex network can be represented as $G^{double} = (G^U, G^L)$. The up layer $G^U = (V^U, E^U, W^U)$ corresponds to the network of DIIs, while the down layer $G^L = (V^L, E^L, W^L)$ represents the network of QFIIs; $V$ is shared by both layers of the network, with $V = \{V_1, V_2, \cdots, V_n\}$, $n$ being the sum of nodes in the network; $E$ is the set of connecting edges $e_{ij}$ between any two nodes $V_i$ and $V_j$ in network $G$, defined as $E = \{e_{ij} = (v_i, v_j)\}$, which signifies that company *i* and *j* share the same QFII or DII. $E^C$ represents the inter-layer edges connecting nodes across different levels in the network (if two companies not only share the same QFII but also have a common DII, then inter-layer edges also exist). $W$ is the set of weights for the undirected graph $G$, representing the magnitude of the connections within and between layers, determined by the number of shared DIIs or QFIIs. $W^C$ represents the number of common QFIIs and DIIs among nodes across different layers, if $w_{ij} = 0$, t indicates that there is no shared equity relationship between companies *i* and *j*.

## 4.3. Variables

(1) The dependent variable: The quality of corporate ESG information disclosure (*Completeness, Suspicion, Inconsistency*).

In light of the current challenges in China's ESG disclosure landscape, this study evaluates the quality of corporate ESG information disclosure through three key dimensions: Completeness, Suspicion, and Inconsistency. These dimensions are chosen for the following reasons:

Firstly, completeness is foundational for assessing a company's ESG performance. The quality of ESG information disclosed is likely to be high only when a company provides comprehensive and exhaustive environmental, social, and governance (ESG) data. Thus, a thorough disclosure serves as the baseline for evaluating the effectiveness of a company's ESG practices.

Secondly, the credibility of a company's ESG disclosure, particularly the consistency between reported performance and actual behavior, is a crucial indicator of information quality in the current Chinese corporate environment. Discrepancies between reported ESG performance and actual actions—commonly referred to as "greenwashing"—undermine the trustworthiness of ESG disclosures. Therefore, ensuring the credibility of ESG reports is key to evaluating their quality.

Lastly, the dimension of Inconsistency reflects the stability and uniformity of ESG ratings across different rating agencies. Large discrepancies in ESG scores for the same company from different agencies could signal inconsistencies in its disclosure practices, which may affect the perceived reliability of the information. The greater the variability between ratings, the higher the likelihood that discrepancies exist in the company's ESG data, thereby diminishing the quality of the disclosure.

Based on these considerations, this study employs the following three indicators to comprehensively assess the quality of ESG information disclosure among firms.

Completeness: This dimension evaluates the thoroughness of a company's social responsibility or annual report in addressing ESG (Environmental, Social, and Governance) concerns comprehensively. Specifically, it scrutinizes the report's disclosure of 10 fundamental elements: "environment and sustainability," "public relations and social welfare," "development and improvement measures for social responsibility systems," "workplace safety," "protection of shareholder rights," "protection of creditor rights," "protection of employee rights," "protection of supplier rights," "protection of customer and consumer rights," and "identification of the company's existing shortcomings." Disclosure for each item is assessed by a scoring system: 1 point for qualitative disclosure, 2 points for quantitative disclosure, and 0 points for non-disclosure. The Completeness index is represented by the natural logarithm of the total score derived from these 10 key elements. A higher index value signifies a more comprehensive disclosure of ESG information, reflecting superior report quality.

Suspicion: Building on the framework proposed by Yu et al. (2020) [28], this study develops an ESG greenwashing score to evaluate the credibility of corporate ESG disclosures. The Bloomberg ESG index quantifies a company's self-reported ESG information, while the Wind ESG rating index assesses actual ESG practices. The Bloomberg ESG Index offers a comprehensive range of environmental, social, and governance data, enabling a thorough evaluation of a company's publicly disclosed ESG performance across areas such as environmental impact, social responsibility, and governance structure. Conversely, Wind ESG Ratings concentrate on quantifying a company's genuine ESG activities by assessing measures in domains like environmental protection, social contributions, and governance practices. Both disclosure and practice scores are standardized to a common scale, and the difference between them is utilized as an indicator of Suspicion. A larger differential suggests a greater likelihood of discrepancies between disclosed ESG information and actual practices, indicating diminished quality and credibility in ESG disclosures. The suspicion index is thus calculated by normalizing the Bloomberg ESG Index and Wind ESG Rating Index onto the same scale and determining the difference between them. Significant discrepancies imply greater inconsistency and reduced trustworthiness of corporate ESG disclosures. The specific calculation formula is as follows:

$$Suspicion_{i,t} = \frac{ESG_{i,t} - \overline{ESG_t}}{\sigma_{ESG_t}} - \frac{Action_{i,t} - \overline{Action_t}}{\sigma_{Action_t}} \tag{2}$$

$\overline{ESG_t}$ represents the average ESG rating index of all companies in year $t$. $\sigma_{ESG_t}$ is the standard deviation of ESG information disclosure for all companies in year $t$. $Action_{i,t}$ indicates the actual ESG action score for company $i$ in year $t$. $\overline{Action_t}$ represents the average actual ESG action score for all companies in year $t$. $\sigma_{Action_t}$ is the standard deviation of actual ESG action scores for all companies in year $t$.

Inconsistency: This study constructs an ESG consistency index following the method proposed by Avramov et al. (2022) [19]. Using ESG rating data from five major domestic and international agencies—Wind, Huazheng, FTSE Russell, SynTao Green Finance, and Menglang—the study ensures comparability across ratings. For each company covered by any two of these agencies, the percentile rank within the sample for each year is calculated and standardized between 0 and 1. The quantile standard deviation (QSD) is then calculated for each company across pairs of rating agencies, resulting in 10 standard deviations for each company (given by $C_5^2$). The average of these 10 standard deviations is used as the measure of ESG information disclosure consistency (Inconsistency). A larger Inconsistency value reflects a greater discrepancy in ESG ratings across agencies, indicating higher uncertainty regarding the quality of the disclosed information.

$$QSD_{it.m,n} = \sqrt{\frac{\left(g_{it.m} - \frac{g_{it.m} + g_{it.n}}{2}\right)^2 + \left(g_{it.n} - \frac{g_{it.m} + g_{it.n}}{2}\right)^2}{2-1}} = \frac{|g_{it.m} - g_{it.n}|}{\sqrt{2}}$$

$$Inconsistency_{i,t} = \frac{QSD_{1,2} + QSD_{1,3} + QSD_{1,4} + \cdots + QSD_{4,5}}{10} \tag{3}$$

$g_{it.m}$ represents the percentile ranking of company $i$ in rating agency $m$ during year $t$, and $QSD_{it.m,n}$ represents the quantile standard deviation for company $i$ under two different ratings during year $t$.

(2) The core independent variable: Network position

In network analysis, methods for measuring node importance predominantly include Degree Centrality, Closeness Centrality, Betweenness Centrality, and Eigenvector Centrality. However, to account for the varying importance of adjacent nodes and to effectively capture the global impact of nodes within the network—especially considering the potential for multi-hop connections between nodes—this study utilizes Eigenvector Centrality as the core explanatory variable. This approach enables a more comprehensive assessment of a company's position within the network, considering not only direct connections but also the influence exerted by those connections across the network.

Eigenvector Centrality goes beyond merely measuring the number of direct connections a node has; it also incorporates the centrality of the nodes to which it is connected. In essence, a node's importance is determined not only by how many important nodes it is linked to but also by the centrality of those connected nodes, thus capturing the global influence of the node in the broader network. This measure is particularly effective in identifying influential nodes situated at the heart of the network.

In the robustness checks section, Katz Centrality is employed as a proxy variable for Eigenvector Centrality. Katz Centrality, similar to Eigenvector Centrality, assesses the importance of a node by considering the influence of its neighbors, their neighbors, and so on. However, Katz Centrality introduces an additional element by assigning greater weight to shorter paths, thus diminishing the impact of more distant nodes. This variation of Eigenvector Centrality takes into account the path length between nodes, enhancing the model by emphasizing proximity in the network. The detailed definition is as follows:

$$EigCent_{i,t} = \lambda \sum_{j=1}^{N_t} E_{i,j,t} \, Eigenvector_{j,t}$$

(4)

$$Katz_{i,t} = \alpha \sum_{j=1}^{N_t} E_{i,j,t} \, Katz_{j,t} + \beta$$

(5)

$N_t$ represents the total number of nodes in the network for year $t$; $\lambda$ is the largest eigenvalue of the adjacency matrix $E_{i,j,t}$. $QFII\_EigCent_{i,t}$ and $DII\_EigCent_{i,t}$ represent the Eigenvector Centrality of the nodes in the QFII layer and DII layer of the complex network respectively. The higher the value of this indicator, the more important the company is within the network layer, indicating a closer position to the network core. This suggests that the company has more access to resources and receives more attention.

(3) Control variables: Following Li et al. (2021) [29], this study includes a series of firm-level control variables to account for other potential factors that could affect the empirical results. These variables include: Return on Assets (*Roa*), Firm Size (*Size*), Equity Balance (*Balance*), Firm Growth (*Growth*), Book-to-Market Ratio (*BM*), Firm Age (*Age*), Cash Flow Level (*Cfl*), The Nature of Property Rights (*SOE*), Debt-to-Asset Ratio (*Lev*). Additionally, considering that interlocking directorships are common in China and can influence internal decision-making and governance, this study controls Board Network Centrality (*DC*) as well. To further account for individual and year-specific factors that might impact the empirical results, the regression analysis also includes firm fixed effects and year fixed effects (detailed description of the variables and the results of the descriptive statistics are provided in Tables 2 and 3). To further prevent possible multicollinearity interference between control variables, we tested for multicollinearity by VIF values before performing benchmark

**Table 2. Definition of Key Variables.**

| Variable types | Variable names | Variable symbols | Variable definitions |
|---|---|---|---|
| Dependent variable | Quality of ESG disclosure | *Completeness* | Ln (Sum of the scores for ESG-related matters disclosed +1) |
| | | *Suspicion* | Corporate ESG disclosure-ESG in action, see the text for details |
| | | *Inconsistency* | Means of interquartile standard deviations across ESG rating agencies |
| Core independent variables | The position in network | *QFII_EigCent* | Eigenvector centrality of firms at different network layers |
| | | *DII_EigCent* | |
| Control variables | Return on Assets | *Roa* | Net profit/Total assets |
| | Debt-to-Asset Ratio | *Lev* | Total debts/ Total assets |
| | Firm Age | *Age* | Years of business operation |
| | Cash Flow Level | *Cfl* | Net operating cash flows/Total assets |
| | Book-to-Market Ratio | *BM* | Net assets/ market capitalization |
| | Firm Growth | *Growth* | Tobin's Q: Market capitalization/Total assets |
| | The Nature of Property Rights | *SOE* | Non-state-owned enterprises = 0, state-owned enterprises = 1 |
| | Firm Size | *Size* | Natural logarithm of total assets |
| | Equity Balance | *Balance* | Proportion of shares held by the second to fifth largest share-holders/ Shareholding of the largest shareholder |
| | Board Network Centrality | *DC* | Degree Centrality |

This table reports the meaning, abbreviations, and calculation methods of the key variables (core explanatory variables, explanatory variables, and control variables) in the empirical part.

**Table 3. Descriptive Statistics.**

| Variables | N | Mean | SD | Min | Max |
|---|---|---|---|---|---|
| *Suspicion* | 6 934 | −0.407 | 1.191 | −3.801 | 5.691 |
| *Inconsistency* | 19 072 | 0.187 | 0.123 | 0.000 | 0.675 |
| *Completeness* | 28 176 | 2.594 | 0.574 | 0.000 | 3.367 |
| *QFII_EigCent* | 37 260 | 0.024 | 0.105 | 0.000 | 1.000 |
| *DII_EigCent* | 37 260 | 0.037 | 0.138 | 0.000 | 1.000 |
| *Roa* | 24 451 | 0.030 | 0.094 | −3.994 | 0.786 |
| *SOE* | 28 173 | 0.319 | 0.466 | 0.000 | 1.000 |
| *Balance* | 28 173 | 0.771 | 0.627 | 0.000 | 4.000 |
| *DC* | 28 173 | 0.415 | 0.357 | 0.000 | 3.000 |
| *Growth* | 22 838 | 2.168 | 0.592 | 0.000 | 3.575 |
| *Lev* | 22 838 | 0.439 | 0.209 | 0.008 | 3.919 |
| *Cfl* | 22 838 | 0.052 | 0.120 | −5.966 | 4.636 |
| *Age* | 22 838 | 11.763 | 7.431 | 2.000 | 32.000 |
| *Size* | 22 838 | 22.370 | 1.324 | 14.940 | 28.640 |
| *BM* | 28 174 | 0.322 | 0.176 | −1.671 | 1.228 |

This table reports descriptive statistical results for the key variables (core explanatory variables, explanatory variables, control variables) in the empirical part. N represents the sample size, Mean represents the mean, SD represents the standard deviation, Min represents the minimum value, and Max represents the maximum value.

regression (the regression results are shown in Table 4). The results showed that the VIF values of the control variables were < 3 and the average VIF was around 1 under the measurement of the three explanatory variables, so it can be considered that there was no multicollinearity problem among the control variables in this study.

### 4.4. Model design

(1) Baseline regression model.

To investigate the impact of the QFII-DII complex network on the quality of corporate ESG information disclosure, this study constructs the following regression model. Hausman test results (with a p-value less than 0.05) suggest the use of the individual time point fixed effect model, which helps account for both individual-level and time-level variations. By selecting a two-way fixed effect model, the regression analysis controls for individual-specific differences as well as time-specific effects. This methodological choice improves the accuracy of the estimation results by reducing omitted variable bias and ensuring that the analysis focuses on the dynamic changes within individual and temporal variations, rather than being influenced by external factors unique to either individual units or time periods:

$$ESG\_Quality_{i,t} = \beta_0 + \beta_1 QFII\_EigCent_{i,t} + \beta_2 \sum Controls_{i,t} + \lambda_i + \gamma_t + \varepsilon_{i,t} \tag{6}$$

$$ESG\_Quality_{i,t} = \beta_0 + \beta_1 DII\_EigCent_{i,t} + \beta_2 \sum Controls_{i,t} + \lambda_i + \gamma_t + \varepsilon_{i,t} \tag{7}$$

Where $ESG\_Quality_{i,t}$ represents the quality of ESG information disclosure for company $i$ in year $t$, including *Completeness, Suspicion* and *Inconsistency*, which measure the completeness, credibility, and consistency of corporate ESG disclosure separately. A higher *Suspicion* or *Inconsistency* value indicates lower credibility and consistency, while a higher *Completeness* value suggests greater disclosure completeness. $DII\_EigCent_{i,t}$ and $QFII\_EigCent_{i,t}$ represent the importance of company $i$ in year $t$ among the Domestic Institutional Investor (DII) network and Qualified Foreign Institutional Investor (QFII) network respectively. *Controls* represent the control variables, $\lambda$ and $\gamma$ denote individual fixed effects and time fixed effects respectively. $\varepsilon$ is the random disturbance term. $\beta_1$ is the key focus of this study, if the coefficient $\beta_1$ is significantly negative for the regression with *Suspicion* and *Inconsistency*, and significantly positive for the regression with

**Table 4. Multicollinearity Test (VIF).**

| Variables | VIF | VIF | VIF |
| --- | --- | --- | --- |
| | (Y = Completeness) | (Y = Suspicion) | (Y = Inconsistency) |
| *Roa* | 1.24 | 1.56 | 1.35 |
| *SOE* | 1.41 | 1.31 | 1.41 |
| *Balance* | 1.07 | 1.08 | 1.07 |
| *DC* | 1.01 | 1.01 | 1.01 |
| *Growth* | 1.07 | 2.02 | 2.12 |
| *Lev* | 2.16 | 2.97 | 2.99 |
| *Cfl* | 1.07 | 1.22 | 1.18 |
| *Age* | 1.41 | 1.21 | 1.40 |
| *Size* | 1.76 | 1.67 | 1.78 |
| *BM* | 1.51 | 2.36 | 2.65 |
| *Mean VIF* | 1.37 | 1.64 | 1.70 |

*Completeness*, it can be inferred that the QFII-DII complex network can promote high-quality ESG development, supporting the theoretical expectations outlined in this paper.

(2) Mechanism Model

To further explore the pathways through which the complex network impacts the quality of corporate ESG information disclosure, this study constructs the following models to test the information exchange effect and market oversight effect:

$$ESG\_Quality_{i,t} = \theta_0 + \theta_1 QFII * DII\_Eig_{i,t} + \theta_2 QFII\_EigCent_{i,t}$$

$$+ \theta_3 DII\_EigCent_{i,t} + \theta_4 \sum Controls_{i,t} + \lambda_i + \gamma_t + \varepsilon_{i,t} \tag{8}$$

$$Attention_{i,t} = \alpha_0 + \alpha_1 QFII\_EigCent_{i,t} + \alpha_2 \sum Controls_{i,t} + \lambda_i + \gamma_t + \varepsilon_{i,t} \tag{9}$$

$$Attention_{i,t} = \alpha_0 + \alpha_1 DII\_EigCent_{i,t} + \alpha_2 \sum Controls_{i,t} + \lambda_i + \gamma_t + \varepsilon_{i,t} \tag{10}$$

In the following, Equation (8) represents the model for testing the information exchange effect, focusing on the coefficient $\theta_1$ of the interaction term $QFII * DII\_Eig_{i,t}$. Equations (9) and (10) are for testing the market oversight effect, where $Attention_{i,t}$ is the analyst coverage data, indicating the level of market attention a company receives. This section focuses on the regression coefficients $\alpha_1$ for the two core explanatory variables.

(3) Heterogeneity model

To test heterogeneity related to the knowledge transfer effect, this study constructs the moderating effect models, represented by Equations (11) and (12), to examine the variation (with DII having a similar structure):

$$ESG\_Quality_{i,t} = \delta_0 + \delta_1 QFII * GA_{i,t} + \delta_2 QFII\_EigCent_{i,t} + \delta_3 GA_{i,t}$$

$$+ \delta_4 \sum Controls_{i,t} + \lambda_i + \gamma_t + \varepsilon_{i,t} \tag{11}$$

$$ESG\_Quality_{i,t} = \delta_0 + \delta_1 QFII * GOV_{i,t} + \delta_2 QFII\_EigCent_{i,t} + \delta_3 GOV_{i,t}$$

$$+ \delta_4 \sum Controls_{i,t} + \lambda_i + \gamma_t + \varepsilon_{i,t} \tag{12}$$

In Equations (11) and (12), which are based on the baseline regression model, additional firm characteristics are included that might affect the network's impact on ESG disclosure quality. Green Awareness (GA) and Internal Governance Level (GOV). The symbols for other variables retain the same meanings as in the baseline regression model. This section focuses on the interaction terms $QFII * GA_{i,t}$, $QFII * GOV_{i,t}$, $DII * GA_{i,t}$, $DII * GOV_{i,t}$ and their coefficient $\delta_1$. If the dependent variables are Suspicion and Inconsistency, and $\delta_1$ is significantly negative, or if the dependent variable is Completeness, and $\delta_1$ is significantly positive, it indicates that improved green awareness and internal governance levels enhance the QFII-DII complex network's effect on the quality of corporate ESG information disclosure.

## 5. Analysis of empirical results

### 5.1. Results of baseline regression model

The results of the baseline regression are presented in Table 5, which reports the effects of a company's network position in the QFII layer (*QFII_EigCent*) and the DII layer (*DII_EigCent*) on the quality of corporate ESG information disclosure (Completeness, *Suspicion*, and *Inconsistency*). Columns (1), (3), and (5) contain the estimates from Model (6),

demonstrating that the regression coefficients for a company's network position in the QFII layer are significant across all three ESG indicators, suggesting that the more central a company is within the QFII network, the higher its ESG information completeness, credibility, and consistency. Columns (2), (4), and (6) represent the estimates from Model (7), indicating that the regression coefficients for the core explanatory variable, DII_EigCent, are significant at the 5% and 1% levels, suggesting that a higher network position in the DII layer also contributes to improved ESG information disclosure quality. Consequently, we can conclude that the QFII-DII complex network promotes high-quality ESG information disclosure, thereby supporting Hypothesis H1.

**Table 5. The Impact of QFII-DII Complex Networks on the Quality of Corporates.**

| Variables | Completeness | | Suspicion | | Inconsistency | |
|---|---|---|---|---|---|---|
| | (1) | (2) | (3) | (4) | (5) | (6) |
| QFII_EigCent | 0.047** | | −0.363*** | | −0.018** | |
| | (2.47) | | (−3.74) | | (−2.47) | |
| DII_EigCent | | 0.040*** | | −0.286*** | | −0.013** |
| | | (2.61) | | (−2.72) | | (−2.29) |
| Roa | 0.037 | 0.036 | 0.611** | 0.618** | −0.024 | −0.024 |
| | (1.07) | (1.06) | (2.10) | (2.13) | (−1.44) | (−1.42) |
| SOE | −0.059** | −0.059** | 0.050 | 0.033 | −0.010 | −0.011 |
| | (−2.39) | (−2.38) | (0.37) | (0.24) | (−1.25) | (−1.27) |
| Balance | −0.002 | −0.002 | 0.066 | 0.062 | 0.002 | 0.001 |
| | (−0.19) | (−0.13) | (0.89) | (0.84) | (0.34) | (0.30) |
| DC | 0.007 | 0.007 | 0.010 | 0.011 | −0.002 | −0.002 |
| | (0.65) | (0.63) | (0.16) | (0.18) | (−0.44) | (−0.42) |
| Growth | 0.002*** | 0.002*** | 0.010 | 0.009 | −0.001 | −0.001 |
| | (7.54) | (7.39) | (0.66) | (0.60) | (−0.46) | (−0.54) |
| Lev | −0.224*** | −0.227*** | 0.088 | 0.107 | −0.011 | −0.011 |
| | (−5.27) | (−5.34) | (0.33) | (0.41) | (−0.64) | (−0.62) |
| Cfl | −0.030 | −0.031 | −0.136 | −0.138 | 0.018* | 0.018** |
| | (−0.76) | (−0.78) | (−1.46) | (−1.47) | (1.95) | (1.97) |
| Age | −0.144*** | −0.144*** | 0.022** | 0.024** | −0.005*** | −0.005*** |
| | (−60.85) | (−60.80) | (1.96) | (2.12) | (−5.78) | (−5.65) |
| Size | 0.070*** | 0.071*** | −0.028 | −0.030 | 0.003 | 0.003 |
| | (7.03) | (7.10) | (−0.46) | (−0.50) | (0.69) | (0.64) |
| BM | −0.074** | −0.074** | −0.144 | −0.163 | −0.020 | −0.021 |
| | (−2.40) | (−2.41) | (−0.70) | (−0.80) | (−1.27) | (−1.34) |
| Constant | 2.324*** | 2.312*** | −0.178 | −0.133 | 0.203** | 0.207** |
| | (11.66) | (11.61) | (−0.14) | (−0.10) | (2.42) | (2.46) |
| N | 22 342 | 22 342 | 6 100 | 6 100 | 16 131 | 16 131 |
| R² | 0.623 | 0.623 | 0.006 | 0.006 | 0.019 | 0.019 |
| Firm & Year FE | Yes | Yes | Yes | Yes | Yes | Yes |

This table reports the baseline regression results. Columns (1), (3), and (5) report the the effects of a company's network position in the QFII layer (QFII_EigCent) on the quality of corporate ESG information disclosure (Completeness, Suspicion, and Inconsistency), Columns (2), (4), and (6) represent the effects of a company's network position in the DII layer (DII_EigCent) on the quality of corporate ESG information disclosure (Completeness, Suspicion, and Inconsistency). All models controlled for time and individual fixed effects (two-way fixed effects). $R^2$ is computed as the squared correlation coefficient between actuals and predicted values. *, ** and *** denote regression coefficients significant at the 10%, 5% and 1% levels respectively. () contains the t-value for a two-tailed test.

## 5.2. Endogenous analysis

This research uses the instrumental variable approach to address potential bidirectional causality. Following Aggarwal et al. (2011) [30], turnover rate (*Turnover*) is selected as the instrumental variable for *QFII_EigCent*. On one hand, QFII is more likely to hold shares of companies with a high turnover rate, which meets the relevance requirement for instrumental variables. On the other hand, the turnover rate is not directly related to ESG disclosure behavior, which satisfies the Exogeneity condition. Based on this, the study uses the turnover rate lagged by one period and applies the Two-Stage Least Squares (2SLS) regression, and the results are presented in Table 6. The second stage (*Second*) results demonstrate that the regression coefficients for *QFII_EigCent* across all three indicators are consistent with the baseline regression, remaining significant at the 10% and 5% levels, which indicates that, after addressing potential endogeneity, the hypotheses can still hold.

Referring to Chen et al. (2020) [31], this study uses the dummy variable *Indexinclu* as an instrumental variable for *DII_EigCent*. A value of 1 is assigned if a stock is included in the Shanghai 180 Index or Shenzhen Component Index, otherwise, it's 0. Such companies often have strong growth and higher investment value, linking them to the company's position in the domestic institutional investor network but not directly to ESG information disclosure quality. Similarly, the

**Table 6. Instrumental Variables Estimation.**

| Variables | First | Second | First | Second | First | Second |
|---|---|---|---|---|---|---|
| | (1) | (2) | (3) | (4) | (5) | (6) |
| | QFII_EigCent | Completeness | QFII_EigCent | Suspicion | QFII_EigCent | Inconsistency |
| L.Turnover | 0.002*** | | 0.005*** | | 0.003*** | |
| | (8.38) | | (4.35) | | (8.55) | |
| QFII_EigCent | | 0.045** | | −5.469** | | −0.352* |
| | | (2.07) | | (−2.08) | | (−1.76) |
| Kleibergen-Paap rk LM statistic | 67.42*** [0.0000] | | 18.25*** [0.0000] | | 70.27*** [0.0000] | |
| Kleibergen-Paap Wald rk F statistic | 70.24 {16.38} | | 18.95 {16.38} | | 73.13 {16.38} | |
| N | 20 007 | 20 007 | 5 333 | 5 333 | 14 947 | 14 947 |
| | DII_EigCent | Completeness | DII_EigCent | Suspicion | DII_EigCent | Inconsistency |
| Indexinclu | 0.196*** | | 0.127*** | | 0.191*** | |
| | (43.89) | | (20.54) | | (42.80) | |
| DII_EigCent | | 0.062*** | | −0.598* | | −0.025** |
| | | (3.89) | | (−1.71) | | (−1.99) |
| Kleibergen-Paap rk LM statistic | 1014.24*** [0.0000] | | 249.19*** [0.0000] | | 951.07*** [0.0000] | |
| Kleibergen-Paap Wald rk F statistic | 1926.27 {16.38} | | 421.99 {16.38} | | 1832.07 {16.38} | |
| N | 22 058 | 22 058 | 6 100 | 6 100 | 15 833 | 15 833 |

This table reports the results of regression based on the instrumental variable method to solve the endogeneity problem. Among them, Turnover and Indexinclu are the instrumental variables selected for QFII and DII. L. Turnover is the lag period variable. Kleibergen-Paap rk LM statistical and Kleibergen-Paap Wald rk F statistical are the results of the unidentifiable and weak instrumental variable tests for instrumental variables. Columns (1), (3) and (5) are the first-stage regression results of the instrumental variable method, and the relationship between the selected instrumental variables and the core explanatory variables is reported. Columns (2), (4) and (6) are the results of the second stage of regression, which report the regression results of the core explanatory variable to the three indicators of the explanatory variable after overcoming the endogeneity problem. *, ** and *** denote regression coefficients significant at the 10%, 5% and 1% levels respectively. () contains the t-value for a two-tailed test. [] contains the p-value, and {} contains the critical value for the weak identification test at the 10% significance level.

second stage results reveal that the direction and significance of the *DII_EigCent* regression coefficient are consistent with the baseline regression, further confirming the robustness of the study's conclusions.

## 5.3. Robustness test

This study conducts robustness test on the baseline conclusions through the following methods:

(1) Replacing the core independent variable: Using Katz centrality (KZ) to measure a company's network position.

(2) Excluding event impact: In response to the 2015 stock market crash, the government introduced certain "national team" shareholders to stabilize the stock market and restore market order, which significantly increased the number of domestic institutional investor relationships. To exclude the impact of this special event, the 2015 data are removed, and the regression is re-run.

(3) Propensity score matching (PSM): A dummy variable is created for each company and year. If it's *EigCent* is greater than the sample median, the value will be 1 classified as the treatment group; otherwise, the value will be 0, indicating it belongs to the control group. Matching is conducted using the nearest neighbor method with a 1:1 ratio, followed by baseline regression on the matched results (The specific results are shown in Table 7).

## 5.4. Mechanism test

(1) Knowledge Transfer Effect: To test related hypotheses, this study refers to the approach by Xie and Lv (2022) [32], based on the core concepts of ESG and international ESG assessment frameworks, using 41 indicators from the World Bank's ESG database and global entropy method to calculate a national ESG index. This provides each company's ESG index for the country of its largest QFII for each year. Following the study by Song et al. (2023) [33], the ESG level of the largest QFII's home country is compared with China's ESG index to determine the relative level of ESG development. If the ESG index for the country of the largest QFII is higher than China's, the ESG_Level is set to 1, indicating that the company is at a knowledge disadvantage; otherwise, it's 0, indicating a knowledge advantage. The group regression results in Table 8 show that in the "knowledge disadvantage" group (i.e., Weak = 1), the regression coefficients for *QFII_EigCent* and *DII_EigCent* are more significant (see Table 8 columns (1)-(6)). This suggests that when companies lack sufficient ESG expertise and management capacity, being in a central position in the network significantly improves the quality of ESG information disclosure (primarily in terms of increased authenticity and completeness). This implies that companies can access more abundant ESG knowledge resources and management experience from abroad through the knowledge transfer effect within the QFII-DII network, supporting Hypothesis H2.

(2) Information Exchange Effect: From the previous conclusions, it is evident that the QFII-DII complex network exhibits a knowledge transfer effect from overseas to China, with further dissemination through the network. On the other hand, QFII, as "outsiders," may lack information on crucial aspects of China's ESG practices, while DII tends to be more familiar with the domestic market environment and ESG practices. This creates a bidirectional knowledge flow where DII provides local information to QFII, fostering complementary and reciprocal information exchange. Based on this, the study explores potential information exchange effects across different network layers. Columns (1), (4), and (7) of Table 9 report the regression results for the sample that exists solely in the QFII network layer (i.e., companies that share QFII with others but lack common DII network resources). The regression coefficients for *QFII_EigCent* are significantly negative at the 1% level for *Suspicion* and *Inconsistency*, but not significant for *Completeness*. This indicates that higher centrality in the QFII layer helps improve the consistency and credibility of ESG information disclosure, but has a less noticeable impact on disclosure completeness. Columns (2), (5), and (8) of Table 8 show the regression results for samples in the DII network layer only. The core explanatory variable *DII_EigCent* has significant regression

 

**Table 7. Robustness Test.**

| Variables | Completeness | | Suspicion | | Inconsistency | |
|---|---|---|---|---|---|---|
| | (1) | (2) | (3) | (4) | (5) | (6) |
| Panel A: Replacing the core independent variable | | | | | | |
| QFII_KZ | 0.005** | | −0.026** | | −0.003*** | |
| | (2.31) | | (−2.30) | | (−2.74) | |
| DII_KZ | | 0.014** | | −0.065** | | −0.004* |
| | | (2.14) | | (−2.04) | | (−1.84) |
| N | 22 204 | 22 204 | 6 063 | 6 063 | 15 995 | 15 995 |
| R² | 0.617 | 0.617 | 0.005 | 0.005 | 0.017 | 0.017 |
| Panel B: Excluding event impact | | | | | | |
| QFII_EigCent | 0.040* | | −0.417*** | | −0.015** | |
| | (1.93) | | (−3.90) | | (−2.02) | |
| DII_EigCent | | 0.043*** | | −0.012** | | −0.251** |
| | | (2.76) | | (−1.99) | | (−2.34) |
| N | 20 511 | 20 511 | 5 535 | 5 535 | 15 330 | 15 330 |
| R² | 0.631 | 0.631 | 0.008 | 0.006 | 0.018 | 0.018 |
| Panel C: Propensity score matching (PSM) | | | | | | |
| QFII_EigCent | 0.038* | | −0.484*** | | −0.028*** | |
| | (1.78) | | (−4.25) | | (−3.17) | |
| DII_EigCent | | 0.032* | | −0.312** | | −0.019*** |
| | | (1.70) | | (−2.57) | | (−3.05) |
| N | 7 975 | 9 229 | 2 068 | 2 870 | 4 585 | 7 540 |
| R² | 0.552 | 0.627 | 0.037 | 0.012 | 0.040 | 0.007 |
| CV | Yes | Yes | Yes | Yes | Yes | Yes |
| Firm &Year FE | Yes | Yes | Yes | Yes | Yes | Yes |

This table reports the results of the robustness test. Panel A is the regression result of Method 1 (substituting the core explanatory variable), Panel B is the regression result of Method 2 (excluding the impact of events), and Panel C is the regression result of Method 3 (propensity score matching PSM). *, ** and *** denote regression coefficients significant at the 10%,5% and 1% levels respectively. () contains the t-value or z-value for a two-tailed test.

coefficients for all three ESG information disclosure quality indicators. However, when compared to the sample in the QFII-only network layer, the promotion effect on ESG information credibility and consistency is relatively weaker. Columns (3), (6), and (9) of Table 8 contain the results for testing the interaction effect. The regression coefficients for the interaction term (QFII * DII_Eig) are significantly positive, indicating that the interaction between different network layers enhances their impact on corporate ESG information disclosure. This validates Hypothesis H3.

(3) Market Oversight Effect: Existing literature often considers the number of analysts following a company as a visual indicator of market attention—the more analysts tracking a company, the higher the market's interest in it [34]. Following Chen et al. (2015) [35], this study uses data of analysts' attention as a proxy variable for market oversight, measured by the natural logarithm of the number of analysts (or teams) tracking the company in a given year plus one. Columns (1) and (2) of Table 10 show that the regression coefficients for QFII_EigCent and DII_EigCent are significantly positive at the 1% level, indicating that a higher network position in both the QFII and DII layers substantially increases analysts' attention, forming a market oversight effect. This validates Hypothesis H4. Additionally, the regression results in Columns (3) through (5) of Table 10 further explore the impact of market oversight on the quality of corporate ESG information disclosure. It shows that the regression coefficient for Attention on Completeness is

 

**Table 8. Knowledge Transfer Effect.**

| Variables | Completeness | | | | Suspicion | | | | Inconsistency | | | |
|---|---|---|---|---|---|---|---|---|---|---|---|---|
| | (1) | (2) | (3) | (4) | (5) | (6) | (7) | (8) | (9) | (10) | (11) | (12) |
| | Weak=1 | Weak=0 | Weak=1 | Weak=0 | Weak=1 | Weak=0 | Weak=1 | Weak=0 | Weak=1 | Weak=0 | Weak=1 | Weak=0 |
| QFII_EigCent | 0.054** | −0.011 | | | −0.470*** | −0.193 | | | −0.024** | −0.008 | | |
| | (2.39) | (−0.28) | | | (−4.11) | (−1.16) | | | (−2.54) | (−0.69) | | |
| DII_EigCent | | | 0.065*** | −0.009 | | | −0.285** | −0.320* | | | −0.019** | −0.005 |
| | | | (3.17) | (−0.40) | | | (−2.13) | (−1.90) | | | (−2.44) | (−0.58) |
| Fisher's Permutation test | 0.065*** {p=0.001} | | 0.074*** {p=0.001} | | 0.277*** {p=0.001} | | −0.035 {p=0.400} | | 0.015 {p=0.400} | | 0.013 {p=0.200} | |
| N | 14 277 | 8 065 | 14 277 | 8 065 | 2 724 | 3 376 | 2 724 | 3 376 | 9 881 | 6 250 | 9 881 | 6 250 |
| R² | 0.657 | 0.565 | 0.657 | 0.565 | 0.014 | 0.012 | 0.010 | 0.014 | 0.022 | 0.022 | 0.021 | 0.022 |
| CV | Yes | Yes | Yes | Yes | Yes | Yes | Yes | Yes | Yes | Yes | Yes | Yes |
| Firm FE | Yes | Yes | Yes | Yes | Yes | Yes | Yes | Yes | Yes | Yes | Yes | Yes |
| Year FE | Yes | Yes | Yes | Yes | Yes | Yes | Yes | Yes | Yes | Yes | Yes | Yes |

The table reports the results of the mechanistic test of the knowledge transfer effect, using the method of group regression. If the ESG index for the country of the largest QFII is higher than China's, the ESG_Level is set to 1 (Weak=1), indicating that the company is at a knowledge disadvantage; otherwise, it's 0 (Weak=0), indicating a knowledge advantage. *, ** and *** denote regression coefficients significant at the 10%,5% and 1% levels respectively. () contains the t-value for a two-tailed test.{}contains the empirical p-value for *Fisher's Permutation test* for inter-group coefficient difference. If the p-value is too small to retain three decimal places, it's marked as 0.001, obtained through 1,000 samples.

**Table 9. Information Exchange Effect Test.**

| Variables | Completeness | | | Suspicion | | | Inconsistency | | |
|---|---|---|---|---|---|---|---|---|---|
| | (1) | (2) | (3) | (4) | (5) | (6) | (7) | (8) | (9) |
| QFII_EigCent | 0.021 | | 0.040** | −0.305*** | | −0.308*** | −0.022*** | | −0.017** |
| | (1.03) | | (2.09) | (−2.67) | | (−3.13) | (−2.71) | | (−2.29) |
| DII_EigCent | | 0.035** | 0.046*** | | −0.229** | −0.258** | | −0.013** | −0.013** |
| | | (2.31) | (2.79) | | (−2.03) | (−2.43) | | (−2.20) | (−2.23) |
| QFII*DII_Eig | | | 0.814*** | | | −31.928*** | | | −0.146** |
| | | | (3.90) | | | (−4.95) | | | (−2.45) |
| N | 17 204 | 17 966 | 22 342 | 4 454 | 4 956 | 6 100 | 11 862 | 13 670 | 16 131 |
| R² | 0.601 | 0.641 | 0.623 | 0.007 | 0.004 | 0.011 | 0.016 | 0.017 | 0.019 |
| CV | Yes | Yes | Yes | Yes | Yes | Yes | Yes | Yes | Yes |
| Firm FE | Yes | Yes | Yes | Yes | Yes | Yes | Yes | Yes | Yes |
| Year FE | Yes | Yes | Yes | Yes | Yes | Yes | Yes | Yes | Yes |

This table reports the results of the mechanism test of the information interaction effect. The regression coefficient of the QFII*DII_Eig interaction term is the focus of attention. *, ** and *** denote regression coefficients significant at the 10%,5% and 1% levels respectively. () contains the t-value or z-value for a two-tailed test.

significantly positive at the 1% level, indicating that analysts' attention mainly contributes to improving the completeness of ESG information disclosure.

## 6. Further analysis

### 6.1. Time-varying characteristics of network effects

The growing openness of China's capital market has promoted increased integration between domestic and foreign capital, reshaping the structure of the QFII-DII composite network. To analyze the time-varying characteristics of this

**Table 10. Market Oversight Effect Test.**

| Variables | Attention | | Quality of ESG information disclosure | | |
|---|---|---|---|---|---|
| | | | Completeness | Suspicion | Inconsistency |
| | (1) | (2) | (3) | (4) | (5) |
| QFII_EigCent | 0.162*** | | | | |
| | (3.40) | | | | |
| DII_EigCent | | 0.069** | | | |
| | | (2.11) | | | |
| Attention | | | 0.018*** | 0.030 | 0.003 |
| | | | (4.38) | (1.10) | (1.45) |
| Constant | −9.961*** | −9.976*** | 2.486*** | 0.077 | 0.230*** |
| | (−20.66) | (−20.67) | (12.06) | (0.06) | (2.67) |
| N | 22 247 | 22 247 | 22 247 | 6 093 | 16 066 |
| $R^2$ | 0.248 | 0.248 | 0.623 | 0.005 | 0.019 |
| CV | Yes | Yes | Yes | Yes | Yes |
| Firm FE | Yes | Yes | Yes | Yes | Yes |
| Year FE | Yes | Yes | Yes | Yes | Yes |

This table reports the results of the mechanism test of the effect of market surveillance. The regression coefficients of QFII_EigCent and DII_EigCent are the focus of attention. Columns (1) and (2) report the regression results of the core explanatory variables to the mechanism variables, and (3), (4) and (5) are supplementary supporting materials to test the relationship between the mechanism variables and the explanatory variables. *, ** and *** denote regression coefficients significant at the 10%,5% and 1% levels respectively. () contains the t-value or z-value for a two-tailed test.

network effect, the study re-ran regressions over five time intervals, revealing distinct trends. Initially, domestic institutional investors (DII) prioritized the Completeness of ESG disclosure, playing a significant role in improving it earlier than foreign investors (QFII). However, over time, QFII gradually addressed its initial lack of attention to Completeness, and by 2018, both DII and QFII had a significant positive impact on enhancing this aspect of disclosure. In terms of improving the Credibility of ESG disclosures, QFII played a pioneering role starting around 2015, while DII began to catch up around 2019, becoming increasingly important in reducing market skepticism. Similarly, QFII initially led efforts to reduce disclosure inconsistencies, but DII soon followed suit. Overall, the results demonstrate a progressive time-based pattern in which the QFII-DII composite network increasingly enhances ESG disclosure quality, driven by the integration of domestic and foreign capital, which has helped align the ESG disclosure standards of both types of institutional investors and move toward greater consensus (see Fig 3–5).

## 6.2. Analysis of the response of various industries to the impact of the QFII-DII complex network

In general, improving a company's network position in both the QFII and DII layers can help enhance its ESG information disclosure quality. Next, this study examines the response of specific industries to the network's effect. According to the National Economic Industry Classification (GB/T4754-2017) published by the State Administration for Market Regulation, this study selects samples from manufacturing, construction, transportation, finance, and real estate industries and performs group regressions by comparing these with other non-industry-specific companies (Ind = 0). Based on Fisher's combined test and the regression results, it is observed that companies in the transportation and real estate industries show no significant improvement in ESG disclosure with increased network position, suggesting that these two industries do not respond noticeably to the QFII-DII complex network's quality enhancement effects. What's more, the regression results for manufacturing, construction, and finance industries show a stronger response to the QFII-DII complex network's improvement effect. For manufacturing and construction, the motivation for ESG disclosure improvement is mainly driven by the DII network layer formed by domestic institutional investors' shared ownership. In contrast, for the finance

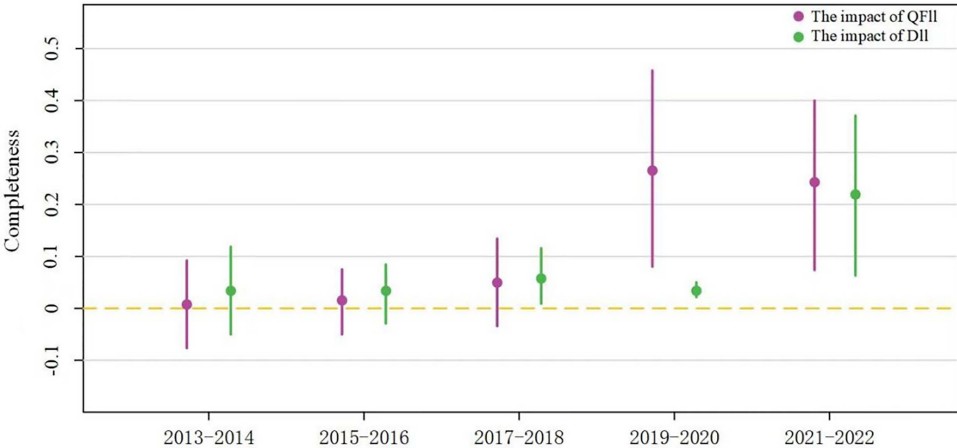

**Fig 3. Visualization of time-phased regression.**

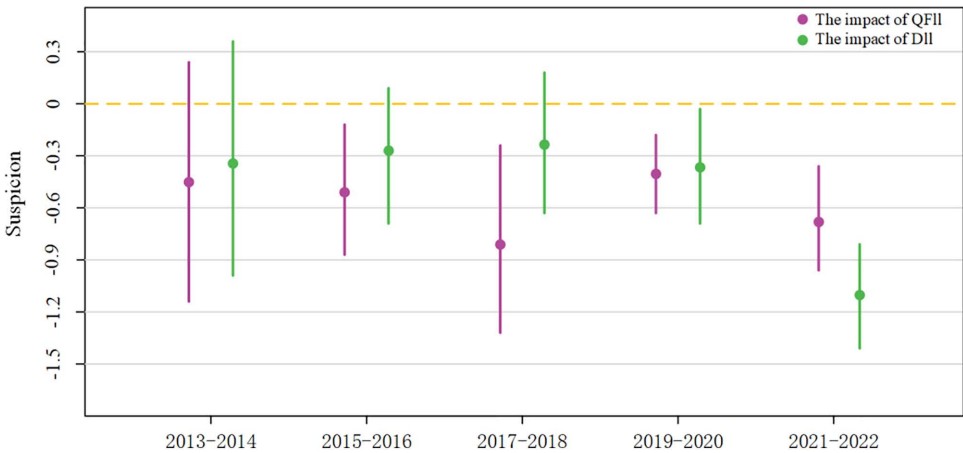

**Fig 4. Visualization of Time-Phased Regression.**

industry, the QFII layer network is also a key factor influencing ESG information disclosure quality. Considering financial globalization and high nature reliance on capital and credit ratings, institutions must pay close attention to foreign capital evaluations and requirements. While foreign capital is also crucial for manufacturing and construction companies, these sectors typically rely more on local markets, focusing on domestic institutional investors' views and support. This is why manufacturing and construction companies respond more strongly to the DII layer network's quality enhancement effects (see Table 11).

### 6.3. Analysis of the impact on institutional investor heterogeneity

Based on the baseline regression and mechanism testing, it's evident that an increase in centrality within the QFII-DII complex network contributes to improving the quality of corporate ESG disclosure, and there is a significant interaction effect between different network layers, which enhances their impact on corporate ESG disclosure. This demonstrates the common ground among various network layers. However, the extended analysis reveals temporal variations in the

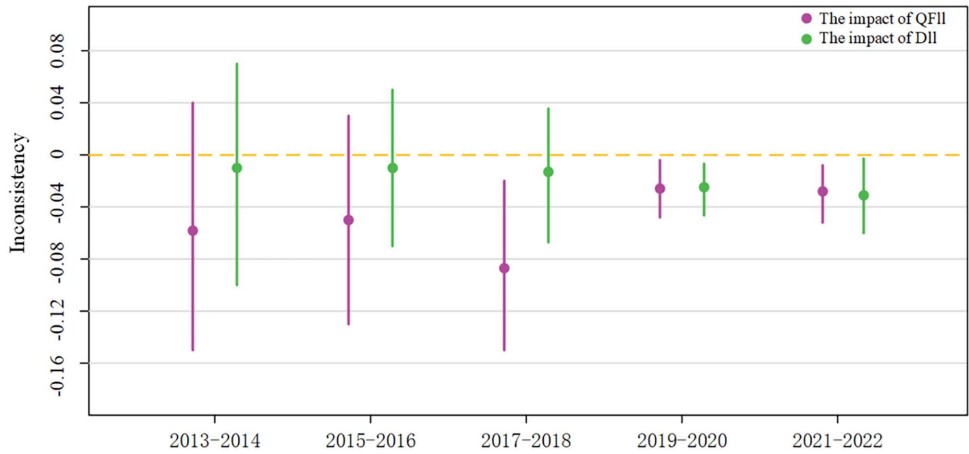

**Fig 5. Visualization of Time-Phased Regression.**

effectiveness of each network layer, with different roles played by the QFII and DII network layers depending on the industry, indicating heterogeneity among network layers.

This leads to a question: does the difference in centrality between QFII and DII network layers for a company impact its ESG disclosure quality? To solve this, the study selects sample companies present in both network layers and calculates the absolute difference in network centrality between DII and QFII layers, which serves as the indicator of *Difference*. The larger the indicator, the greater the heterogeneity between layers, indicating that the company plays a more central role in one layer and is more peripheral in the other. A smaller indicator implies a relatively consistent role and importance across both layers. The sample is divided into DII-preferred groups (Prefer = 1) and QFII-preferred groups (Prefer = 0) according to the sign of the centrality difference, allowing for analysis of the impact of different types of institutional investors. Table 12 shows that when a company's centrality in the DII layer is significantly greater than in the QFII layer, and the centrality difference is larger, the company is more likely to engage in ESG greenwashing. This suggests that companies with more focus from domestic institutional investors than QFII may still face significant issues with ESG disclosure authenticity. It implies that domestic institutional investors might be less rigorous in assessing the authenticity of corporate ESG disclosure compared to foreign institutional investors. This leads to the conclusion that further opening the capital market to high-quality foreign institutional investors is crucial for strengthening China's ESG system. It is through the collaborative effort of domestic and foreign capital that high-quality ESG development can be better promoted.

## 7. Conclusions and implications

This study examines Chinese listed companies from 2013 to 2022, constructing a two-layer complex network of Qualified Foreign Institutional Investors (QFII) and Domestic Institutional Investors (DII) based on significant shareholding information. The research assesses the credibility, consistency, and completeness of ESG information disclosure and investigates how the QFII-DII complex network influences this quality. The findings indicate that a company's centrality within this network positively impacts its ESG disclosure quality.

The study identifies three primary mechanisms through which the QFII-DII network enhances corporate ESG information disclosure quality among Chinese listed companies. First, the Knowledge Transfer Effect enables the dissemination of international ESG expertise and standards, allowing companies positioned at the network's core to access valuable knowledge resources, thereby improving the credibility, consistency, and completeness of their ESG disclosures. Second, the Information Exchange Effect sees QFII and DII acting as intermediaries, fostering information exchange that deepens

**Table 11. Results of Response of Various Industries to the Impact of the QFII-DII Complex Network.**

| Variables | Completeness | | | | Suspicion | | | | Inconsistency | | | |
|---|---|---|---|---|---|---|---|---|---|---|---|---|
| | (1) | (2) | (3) | (4) | (5) | (6) | (7) | (8) | (9) | (10) | (11) | (12) |
| | Ind=1 | Ind=0 | Ind=1 | Ind=0 | Ind=1 | Ind=0 | Ind=1 | Ind=0 | Ind=1 | Ind=0 | Ind=1 | Ind=0 |
| **Panel A: Whether the Company Belongs to the Manufacturing Industry** | | | | | | | | | | | | |
| QFII_EigCent | 0.038* | 0.049 | | | −0.416*** | −0.227 | | | −0.019** | −0.012 | | |
| | (1.79) | (1.35) | | | (−3.72) | (−1.29) | | | (−2.23) | (−0.87) | | |
| DII_EigCent | | | 0.040** | 0.046* | | | −0.304** | −0.227 | | | −0.032*** | −0.006 |
| | | | (2.19) | (1.70) | | | (−2.36) | (−1.20) | | | (−3.32) | (−0.79) |
| Fisher's Permutation test | 0.011 {p=0.343} | | 0.007 {p=0.426} | | 0.189 {p=0.308} | | 0.076 {p=0.430} | | 0.007 {p=0.331} | | 0.026** {p=0.021} | |
| **Panel B: Whether the Company Belongs to the Construction Industry** | | | | | | | | | | | | |
| QFII_EigCent | 0.046** | 0.0373 | | | −0.603* | −0.358*** | | | −0.018** | −0.030 | | |
| | (2.31) | (0.39) | | | (−1.72) | (−3.55) | | | (−2.36) | (−1.09) | | |
| DII_EigCent | | | 0.041*** | −0.028 | | | −0.832** | −0.264** | | | −0.012** | −0.048 |
| | | | (2.67) | (−0.28) | | | (−2.38) | (−2.44) | | | (−2.03) | (−1.47) |
| Fisher's Permutation test | −0.008 {p=0.375} | | −0.069** {p=0.049} | | 0.244 {p=0.127} | | 0.567** {p=0.010} | | −0.012 {p=0.269} | | −0.035*** {p=0.006} | |
| **Panel C: Whether the Company Belongs to the Transportation Industry** | | | | | | | | | | | | |
| QFII_EigCent | 0.145 | 0.047** | | | −1.366 | −0.357*** | | | −0.041 | −0.018** | | |
| | (1.06) | (2.41) | | | (−1.50) | (−3.66) | | | (−0.30) | (−2.45) | | |
| DII_EigCent | | | 0.100 | 0.036** | | | −0.577** | −0.252** | | | −0.015 | −0.013** |
| | | | (1.50) | (2.29) | | | (−2.28) | (−2.23) | | | (−0.73) | (−2.07) |
| Fisher's Permutation test | −0.098* {p=0.051} | | −0.064* {p=0.066} | | 1.009*** {p=0.001} | | 0.324 {p=0.100} | | 0.022 {p=0.118} | | 0.002 {p=0.386} | |
| **Panel D: Whether the Company Belongs to the Finance Industry** | | | | | | | | | | | | |
| QFII_EigCent | 0.117 | 0.066*** | | | −1.392** | −0.394*** | | | −0.016** | 0.0289 | | |
| | (1.04) | (3.77) | | | (−2.44) | (−4.03) | | | (−2.18) | (0.61) | | |
| DII_EigCent | | | 0.078 | 0.038*** | | | −0.283*** | −0.087 | | | −0.099** | −0.008 |
| | | | (0.87) | (3.00) | | | (−2.84) | (−0.17) | | | (−2.51) | (−1.41) |
| Fisher's Permutation test | −0.051 {p=0.415} | | −0.040 {p=0.338} | | 0.998*** {p=0.001} | | 0.196 {p=0.174} | | 0.045** {p=0.046} | | 0.091*** {p=0.001} | |
| **Panel E: Whether the Company Belongs to the Real Estate Industry** | | | | | | | | | | | | |
| QFII_EigCent | 0.027 | 0.048** | | | −0.272 | −0.362*** | | | 0.034 | −0.019*** | | |
| | (0.27) | (2.46) | | | (−0.27) | (−3.74) | | | (0.37) | (−2.63) | | |
| DII_EigCent | | | −0.018 | 0.041*** | | | 0.805 | −0.315*** | | | −0.045 | −0.012** |
| | | | (−0.18) | (2.67) | | | (1.04) | (−2.99) | | | (−1.11) | (−2.08) |
| Fisher's Permutation test | 0.021 {p=0.286} | | 0.059* {p=0.077} | | −0.090 {p=0.338} | | −1.119*** {p=0.001} | | −0.053*** {p=0.001} | | 0.032** {p=0.010} | |

This table reports the results of heterogeneous responses across industries to the impact of complex QFII-DII networks. Panel ABCDE selects samples from the manufacturing, construction, transportation, finance, and real estate industries, respectively, and performs grouped regression by comparing these samples (Ind=1) with the remaining non-industry-specific samples (Ind=0). *, ** and *** denote regression coefficients significant at the 10%,5% and 1% levels respectively. () contains the t-value for a two-tailed test.{}contains the empirical p-value for *Fisher's Permutation test* for inter-group coefficient difference. If the p-value is too small to retain three decimal places, it's marked as 0.001, obtained through 1,000 samples.

**Table 12. Analysis of Inter-Layer Heterogeneity Related to Types of Institutional Investors.**

| Variables | Completeness | | Suspicion | | Inconsistency | |
|---|---|---|---|---|---|---|
| | Prefer=1 | Prefer=0 | Prefer=1 | Prefer=0 | Prefer=1 | Prefer=0 |
| Difference | −0.111 | 0.023 | 4.994*** | 0.243 | 0.036 | 0.001 |
| | (−0.93) | (0.29) | (5.41) | (1.11) | (0.59) | (0.05) |
| Constant | −8.058** | 4.777*** | 24.526** | 16.010* | −0.979 | 2.237** |
| | (−2.50) | (3.18) | (2.47) | (1.89) | (−0.44) | (2.28) |
| Chow Test (P-Value) | 0.954 | | 0.032** | | 0.623 | |
| N | 303 | 343 | 97 | 126 | 183 | 215 |
| $R^2$ | 0.787 | 0.585 | 0.950 | 0.664 | 0.477 | 0.448 |
| CV | Yes | Yes | Yes | Yes | Yes | Yes |
| Firm FE | Yes | Yes | Yes | Yes | Yes | Yes |
| Year FE | Yes | Yes | Yes | Yes | Yes | Yes |

This table reports the results of the analysis of interlayer heterogeneity related to the type of institutional investor. Based on the differences in the centrality of firms in the two-tier network, the sample is divided into DII priority group (Prefer=1) and QFII priority group (Prefer=0), so that the impact of different types of institutional investors can be analyzed. *, ** and *** denote regression coefficients significant at the 10%,5% and 1% levels respectively. () contains the t-value for a two-tailed test. The p-value for coefficient difference is calculated based on the estimation results of the Chow test in the interaction model.

mutual understanding of markets and promotes the convergence of ESG standards, leading to better disclosure practices. Finally, the Market Oversight Effect highlights how companies with greater centrality in the network attract increased market attention, encouraging them to enhance their ESG disclosure quality to meet investor and stakeholder expectations. These mechanisms collectively drive the improvement of ESG disclosure quality among Chinese firms.

Heterogeneity analysis reveals that the positive impact of the QFII-DII network on ESG disclosure quality is more pronounced in companies with higher green awareness and robust internal governance. Additionally, companies audited by the Big Four accounting firms and those in heavily polluting industries experience a more significant improvement in ESG disclosure quality as their position within the network strengthens. The study also observes a time-varying pattern in the convergence of the improvement effect across different network layers.

Industry-specific responses vary, with manufacturing, construction, and finance sectors showing stronger reactions to the quality enhancement effects of the QFII-DII network. Manufacturing and construction sectors emphasize domestic institutional investors' perspectives, while the finance sector is more responsive to evaluations from foreign capital. Furthermore, inter-layer heterogeneity among institutional investor networks influences corporate ESG disclosure behavior, with companies exhibiting greater centrality in the DII layer and a larger disparity compared to their centrality in the QFII layer being more prone to ESG greenwashing.

In summary, this study underscores the importance of institutional investors in enhancing ESG information disclosure quality among Chinese listed companies. By improving their position within the QFII-DII complex network and focusing on knowledge transfer, information exchange, and market oversight, companies can achieve higher-quality ESG disclosures, thereby enhancing their global competitiveness and contributing to sustainable development.

Based on the findings of this study, the following policy implications are proposed.

(1) Promoting High-Quality ESG Development through Capital Market Opening. This study highlights the pivotal role of Qualified Foreign Institutional Investors (QFIIs) in transferring international ESG knowledge and promoting high-quality ESG practices in Chinese firms. In line with China's strategy of expanding its capital market opening, policymakers should focus on further enhancing the appeal of the Chinese market to QFIIs. Specific measures include lowering the investment threshold for QFIIs and offering more attractive tax incentives. These steps can attract more QFII

participation, thereby strengthening their capacity to influence ESG practices in the domestic market. In addition, policymakers should encourage QFIIs to actively contribute to the formulation of ESG disclosure standards, which would help standardize the ESG reporting processes among Chinese enterprises, fostering greater transparency and alignment with global best practices.

(2) Leveraging Internal and External Audits for Improved ESG Disclosure. This study reveals that the positive influence of the QFII-DII network on ESG disclosure quality is more evident in firms audited by Big Four accounting firms, emphasizing the crucial role of external audits in ensuring the credibility and accuracy of ESG reporting. To strengthen ESG disclosure standards, policymakers should consider requiring the involvement of reputable third-party auditors in ESG assessments. At the same time, companies should establish dedicated internal audit teams with deep knowledge of their operations to oversee the collection and verification of ESG data. A collaborative approach between internal and external auditors would ensure the high quality of ESG information disclosure, strengthening investor trust and public confidence.

(3) Driving ESG Development through Talent Acquisition and Training. This study finds that companies with senior executives who possess a higher level of ESG awareness tend to disclose ESG information more effectively. This finding underscores the importance of having executives who understand and prioritize ESG factors. To foster high-quality ESG practices, companies should prioritize the recruitment of senior executives with a strong background in sustainability and ESG. Furthermore, enterprises should design targeted ESG training programs for their executives, covering critical aspects such as sustainability trends, regulatory developments, and best practices. Encouraging participation in ESG conferences and workshops can help executives stay current. Integrating ESG performance into executive evaluations can further align leadership incentives with sustainability goals. Ultimately, embedding ESG principles into corporate leadership and decision-making processes is essential for fostering long-term responsible business practices.

This study acknowledges several limitations. First, the network analysis used may introduce some noise by potentially overestimating the dynamic and relational characteristics of the QFII-DII network. Second, the completeness index focuses on the breadth of disclosure but does not capture the depth or accuracy of the information, which may result in low-quality disclosures despite full coverage. Third, the suspicion indicator, based on ratings from Bloomberg and Wind, be affected by subjective differences in their evaluation criteria. Similarly, the Consistency index assumes that discrepancies between rating agencies stem solely from disclosure inconsistencies, but such differences could also arise from variations in agencies' standards or data sources, which may not fully represent the actual quality of ESG disclosures.

## Supporting information

**S1 Data. Data.**
(XLSX)

## Author contributions

**Conceptualization:** Wei Yin.

**Formal analysis:** Wei Yin.

**Methodology:** Yang Su, Ruizhe Wang.

**Software:** Yang Su.

**Writing – original draft:** Berna Kirkulak-Uludag.

**Writing – review & editing:** Yang Su, An Yan.

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
