## [Decision Letter · Decision Letter 0]

25 Jun 2025

PONE-D-25-15484Impact of Domestic and Foreign Investors on ESG Disclosure Quality in Chinese Listed Firms: Divergence or Convergence?PLOS ONE

Dear Dr. Yin,

Thank you for submitting your manuscript to PLOS ONE. After careful consideration, we feel that it has merit but does not fully meet PLOS ONE’s publication criteria as it currently stands. Therefore, we invite you to submit a revised version of the manuscript that addresses the points raised during the review process. 

1. Major revision is required as per the reviewer reports.

2. The article writing structure need to be improved to confirm with PLOS One article style.

Please refer to PLOS One (Finance articles) available at: https://journals.plos.org/plosone/browse/finance

**Comments from the Editorial Office** : In order to comply fully with PLOS One's publication criteria, we have identified the following additional issues that need to be addressed in your revision: 

1. We have noted that in the section titled 'Theories and Hypotheses', several statements have been made without appropriate referencing provided, for example in Pg 12 and 13 and it is further unclear how the random forest analysis on Pg 12 was carried out. Therefore, please revise these sections thoroughly to provide adequate references for the statements, and provide a more detailed description of how the random forest analysis was carried out to comply with PLOS One's publication criteria 3 and 4 which requires that experiments and analyses must be conducted rigorously, with appropriate controls; methods must be described in sufficient detail for others to replicate the analyses; and conclusions must be supported by the results presented (http://journals.plos.org/plosone/s/criteria-for-publication)

2. Further, for this manuscript, we noted that Reviewer 1 has raised concerns regarding the contribution of this study and we have identified several related published works which have not been cited or discussed for example: /10.1016/j.irfa.2025.104032; 10.3390/su16188238. Therefore, please revise your manuscript to clearly discuss the contribution of this study in light of the other related works which have been cited above. 

We look forward to receiving your revised manuscript.

Kind regards,

Jasman Tuyon, Ph.D., MBA

Academic Editor

PLOS One

**Journal Requirements:**

1. When submitting your revision, we need you to address these additional requirements. Please ensure that your manuscript meets PLOS ONE's style requirements, including those for file naming. The PLOS ONE style templates can be found at https://journals.plos.org/plosone/s/file?id=wjVg/PLOSOne_formatting_sample_main_body.pdf and https://journals.plos.org/plosone/s/file?id=ba62/PLOSOne_formatting_sample_title_authors_affiliations.pdf 2. Thank you for stating in your Funding Statement: This work was supported by National Social Science of China Program (24BJY093). the laboratory of Philosophy and Social Sciences at Universities in Jiangsu Province-Fintech and Big Data Laboratory of Southeast University (7514009282).  Please provide an amended statement that declares *all* the funding or sources of support (whether external or internal to your organization) received during this study, as detailed online in our guide for authors at http://journals.plos.org/plosone/s/submit-now.  Please also include the statement “There was no additional external funding received for this study.” in your updated Funding Statement. Please include your amended Funding Statement within your cover letter. We will change the online submission form on your behalf. 3. Thank you for uploading your study's underlying data set. Unfortunately, the repository you have noted in your Data Availability statement does not qualify as an acceptable data repository according to PLOS's standards. At this time, please upload the minimal data set necessary to replicate your study's findings to a stable, public repository (such as figshare or Dryad) and provide us with the relevant URLs, DOIs, or accession numbers that may be used to access these data. For a list of recommended repositories and additional information on PLOS standards for data deposition, please see https://journals.plos.org/plosone/s/recommended-repositories. 4. When completing the data availability statement of the submission form, you indicated that you will make your data available on acceptance. We strongly recommend all authors decide on a data sharing plan before acceptance, as the process can be lengthy and hold up publication timelines. Please note that, though access restrictions are acceptable now, your entire data will need to be made freely accessible if your manuscript is accepted for publication. This policy applies to all data except where public deposition would breach compliance with the protocol approved by your research ethics board. If you are unable to adhere to our open data policy, please kindly revise your statement to explain your reasoning and we will seek the editor's input on an exemption. Please be assured that, once you have provided your new statement, the assessment of your exemption will not hold up the peer review process.

**Additional Editor Comments:**

The article writing structure need to be improved to confirm with PLOS One article style.

Please refer to PLOS One (Finance articles) available at: https://journals.plos.org/plosone/browse/finance

Reviewers' comments:

Reviewer's Responses to Questions

**Comments to the Author**

1. Is the manuscript technically sound, and do the data support the conclusions?

Reviewer #1: Yes

Reviewer #2: Yes

2. Has the statistical analysis been performed appropriately and rigorously? 

Reviewer #1: Yes

Reviewer #2: Yes

3. Have the authors made all data underlying the findings in their manuscript fully available?

Reviewer #1: Yes

Reviewer #2: Yes

4. Is the manuscript presented in an intelligible fashion and written in standard English?

Reviewer #1: Yes

Reviewer #2: Yes

5. Review Comments to the Author

**Reviewer #1:**  Thank you for allowing me to review this excellent manuscript. I respect and appreciate the author's efforts to provide a high-quality manuscript. However, the author may consider the following issues to improve the manuscript.

I. Introduction & Hypothesis Section

1. The literature review needs stronger synthesis. Instead of just listing studies, explicitly connect them, highlighting areas of agreement, disagreement, and gaps in the existing research.

2. Specifically address the limitations of previous ESG research. What aspects have been neglected, and how does this study address those shortcomings?

3. Be critical of the chosen measures for QFII and DII behavior. Are they comprehensive or do they have limitations?

4. More clearly articulate the theoretical mechanisms linking the QFII-DII network to ESG disclosure quality. The diagrams are helpful, but explicitly state which theories (e.g., agency theory, signaling theory, stakeholder theory) underpin your hypotheses.

5. For each hypothesis, provide a more rigorous theoretical justification. Why exactly would a stronger network position lead to better ESG disclosure quality? What are the theoretical assumptions?

II. Research Methods

1. Variable Definitions:

- Provide more detailed justifications for the choice of variables. Why were these specific measures chosen for ESG quality (completeness, suspicion, inconsistency)? What are their limitations? Are there alternative measures that could have been used?

- Explain the rationale behind using the natural logarithm of the sum of scores for the completeness measure, and what is the specific meaning behind these scores?

- Explain the specific ESG indices, databases, and measures used. For example, which Bloomberg and Wind ESG indices are used? How are they constructed?

2. Network Analysis:

- The explanation of network analysis could be more accessible to readers unfamiliar with the technique. Explain the intuitive meaning of Eigenvector Centrality and Katz Centrality in the context of this research.

- Explain why the other centrality measures were not used.

- Address the potential for endogeneity in the network structure. Could ESG disclosure quality itself influence the formation of the QFII-DII network? If so, how is this addressed?

3. Model Specification:

- Justify the choice of fixed effects. Why firm fixed effects and year fixed effects? What are the implications of this choice?

- Address potential multicollinearity issues among the control variables.

4. Limitations: Explicitly acknowledge the limitations of the research design. What are the potential biases or weaknesses of the study? How might these limitations affect the generalizability of the findings?

III. Implications

1. Be more specific about the practical implications of the findings. For whom are these findings relevant (e.g., policymakers, investors, companies)?

2. Avoid overly broad or vague statements.

3. Based on the findings, provide concrete and actionable policy recommendations. For example, what steps can policymakers take to encourage higher-quality ESG disclosure in China?

4. Explain how companies can use the findings to improve their own ESG disclosure practices. Should they focus on attracting certain types of investors? Should they prioritize certain aspects of ESG disclosure?

5. Reiterate how the findings contribute to the existing theoretical understanding of ESG disclosure and institutional investor behavior.

6. Relate the findings to broader debates or issues in the field of corporate governance and sustainability.

I hope these suggestions will help you improve your manuscript. Good luck with your manuscript.

**Reviewer #2:**  Firstly, I strongly suggest that in the exposition of the research hypotheses, it is important to focus on the theoretical deductions and based on the theoretical deductions. Secondly, I think there is a need to enhance the depth of the study, mechanism research, and enhance a more refined.

6. PLOS authors have the option to publish the peer review history of their article (what does this mean? ). If published, this will include your full peer review and any attached files.

**Do you want your identity to be public for this peer review?** For information about this choice, including consent withdrawal, please see our Privacy Policy .

Reviewer #1: No

Reviewer #2: No

---

## [Author Response · Author response to Decision Letter 1]

21 Jul 2025

Response to editor and reviewers

We sincerely thank the editor and reviewers for the constructive comments and suggestions. The comments have been helpful and we believe that in addressing them we have been able to improve the contribution of our manuscript.

Comments from the Editorial Office: In order to comply fully with PLOS One's publication criteria, we have identified the following additional issues that need to be addressed in your revision:

1. We have noted that in the section titled 'Theories and Hypotheses', several statements have been made without appropriate referencing provided, for example in Pg 12 and 13 and it is further unclear how the random forest analysis on Pg 12 was carried out. Therefore, please revise these sections thoroughly to provide adequate references for the statements, and provide a more detailed description of how the random forest analysis was carried out to comply with PLOS One's publication criteria 3 and 4 which requires that experiments and analyses must be conducted rigorously, with appropriate controls; methods must be described in sufficient detail for others to replicate the analyses; and conclusions must be supported by the results presented (http://journals.plos.org/plosone/s/criteria-for-publication)

Answer: We would like to express our gratitude to the editor for bringing this issue to our attention. We have significantly revised the methodology section to include a comprehensive and transparent description of how the random forest analysis was carried out. We have carefully reviewed the section and added appropriate scholarly references to support all theoretical claims and statements that previously lacked citations.

2. Further, for this manuscript, we noted that Reviewer 1 has raised concerns regarding the contribution of this study and we have identified several related published works which have not been cited or discussed for example: /10.1016/j.irfa.2025.104032; 10.3390/su16188238. Therefore, please revise your manuscript to clearly discuss the contribution of this study in light of the other related works which have been cited above.

Answer: We thank the editor and Reviewer 1 for pointing out the need to better articulate the contribution of our study and to engage more thoroughly with the relevant literature. We have added relevant papers, including the two mentioned above, and rewritten the literature review section.

Reviewer #1: Thank you for allowing me to review this excellent manuscript. I respect and appreciate the author's efforts to provide a high-quality manuscript. However, the author may consider the following issues to improve the manuscript.

I. Introduction & Hypothesis Section

1. The literature review needs stronger synthesis. Instead of just listing studies, explicitly connect them, highlighting areas of agreement, disagreement, and gaps in the existing research.

Answer: We sincerely appreciate the reviewer’s constructive comment. In response, we have added a literature review section and revised the introduction to address the identified weaknesses.

2. Specifically address the limitations of previous ESG research. What aspects have been neglected, and how does this study address those shortcomings?

Answer: We thank the reviewer for this valuable suggestion. We have discussed this point in the final paragraph of the literature review section.

3. Be critical of the chosen measures for QFII and DII behavior. Are they comprehensive or do they have limitations?

Answer: We thank the reviewer for pointing this out. The comprehensive discussion has been specifically added at the beginning of Section 4.2, and the limitations are addressed at the end of the paper.

4. More clearly articulate the theoretical mechanisms linking the QFII-DII network to ESG disclosure quality. The diagrams are helpful, but explicitly state which theories (e.g., agency theory, signaling theory, stakeholder theory) underpin your hypotheses.

Answer: We thank the reviewer for the insightful suggestion to more clearly articulate the theoretical foundations linking the QFII-DII network to ESG disclosure quality. In response, we have revised the Hypotheses Section to explicitly incorporate relevant theories underpinning each proposed mechanism

5. For each hypothesis, provide a more rigorous theoretical justification. Why exactly would a stronger network position lead to better ESG disclosure quality? What are the theoretical assumptions?

Answer: We sincerely thank the reviewer for the valuable suggestion to provide a more rigorous theoretical justification for each hypothesis (Stakeholder, Information exchange, and Signaling theories). In the revised manuscript, we have strengthened the theoretical reasoning for all four hypotheses by explicitly articulating why a stronger network position (i.e., higher centrality in the QFII-DII network) is expected to lead to better ESG disclosure quality. The revisions are grounded in well-established theories and clarify the assumptions that support each causal pathway.

II. Research Methods

1. Variable Definitions:

- Provide more detailed justifications for the choice of variables. Why were these specific measures chosen for ESG quality (completeness, suspicion, inconsistency)? What are their limitations? Are there alternative measures that could have been used?

- Explain the rationale behind using the natural logarithm of the sum of scores for the completeness measure, and what is the specific meaning behind these scores?

- Explain the specific ESG indices, databases, and measures used. For example, which Bloomberg and Wind ESG indices are used? How are they constructed?

Answer: We appreciate the reviewer’s valuable comments and suggestions. Section 4.3 has been significantly revised, and we have provided detailed definitions of the variables.

2. Network Analysis:

- The explanation of network analysis could be more accessible to readers unfamiliar with the technique. Explain the intuitive meaning of Eigenvector Centrality and Katz Centrality in the context of this research.

- Explain why the other centrality measures were not used.

- Address the potential for endogeneity in the network structure. Could ESG disclosure quality itself influence the formation of the QFII-DII network? If so, how is this addressed?

Answer: We sincerely thank the reviewer for this constructive comment. We have added specific content to further elaborate on the network analysis, particularly in the section titled “The Core Independent Variable: Network Position.”

3. Model Specification:

- Justify the choice of fixed effects. Why firm fixed effects and year fixed effects? What are the implications of this choice?

- Address potential multicollinearity issues among the control variables.

Answer: We sincerely thank the reviewer for pointing this out. A further explanation has been added in Section 4.3 and 4.4, covering the model design. We added a VIF test to investigate the potential multicollinearity between control variables (the regression results are shown in Table 4). The results show that the average VIF value of the control variables is < 3, and the average VIF is also around 1, so it can be considered that there is no multicollinearity problem among the control variables in this study.

4. Limitations: Explicitly acknowledge the limitations of the research design. What are the potential biases or weaknesses of the study? How might these limitations affect the generalizability of the findings?

Answer: We appreciate the reviewer’s valuable suggestion to explicitly acknowledge the limitations of our research design. In the revised manuscript, we have added a dedicated section at the end of the conclusion section, discussing the potential biases and weaknesses, as well as their implications for the generalizability of our finding.

III. Implications

1. Be more specific about the practical implications of the findings. For whom are these findings relevant (e.g., policymakers, investors, companies)?

2. Avoid overly broad or vague statements.

3. Based on the findings, provide concrete and actionable policy recommendations. For example, what steps can policymakers take to encourage higher-quality ESG disclosure in China?

4. Explain how companies can use the findings to improve their own ESG disclosure practices. Should they focus on attracting certain types of investors? Should they prioritize certain aspects of ESG disclosure?

5. Reiterate how the findings contribute to the existing theoretical understanding of ESG disclosure and institutional investor behavior.

6. Relate the findings to broader debates or issues in the field of corporate governance and sustainability.

Answer: We appreciate the reviewers’ insightful comments, which have helped us strengthen the practical relevance and rigor of our study. We have added a dedicated section on implications, thoroughly developed in response to the reviewer’s valuable suggestions, to clearly outline the practical relevance and more specific practical implications.

Reviewer #2: Firstly, I strongly suggest that in the exposition of the research hypotheses, it is important to focus on the theoretical deductions and based on the theoretical deductions. Secondly, I think there is a need to enhance the depth of the study, mechanism research, and enhance a more refined.

Answer: We appreciate the reviewer’s valuable feedback.In response, we have thoroughly revised the “Hypotheses Section” to clearly derive each hypothesis from established theoretical frameworks such as stakeholder theory, signaling theory, information exchange theory, and institutional theory. We explicitly outline the logical steps connecting theory to expected empirical outcomes.Additionally, we have expanded the discussion of the mechanisms through which the QFII-DII network influences ESG disclosure quality.

These improvements provide a stronger conceptual foundation and enhance the depth of our study as recommended. We appreciate the reviewer’s insightful comments, which have been invaluable in advancing the quality of our manuscript.

---

## [Decision Letter · Decision Letter 1]

1 Aug 2025

Impact of Domestic and Foreign Investors on ESG Disclosure Quality in Chinese Listed Firms: Divergence or Convergence?

PONE-D-25-15484R1

Dear Dr. Yin,

We’re pleased to inform you that your manuscript has been judged scientifically suitable for publication and will be formally accepted for publication once it meets all outstanding technical requirements.

Kind regards,

Jasman Tuyon, Ph.D., MBA

Academic Editor

PLOS ONE

Additional Editor Comments (optional):

Reviewers' comments:

Reviewer's Responses to Questions

**Comments to the Author**

1. If the authors have adequately addressed your comments raised in a previous round of review and you feel that this manuscript is now acceptable for publication, you may indicate that here to bypass the “Comments to the Author” section, enter your conflict of interest statement in the “Confidential to Editor” section, and submit your "Accept" recommendation.

Reviewer #1: All comments have been addressed

Reviewer #2: (No Response)

2. Is the manuscript technically sound, and do the data support the conclusions?

Reviewer #1: Yes

Reviewer #2: Yes

3. Has the statistical analysis been performed appropriately and rigorously? 

Reviewer #1: Yes

Reviewer #2: Yes

4. Have the authors made all data underlying the findings in their manuscript fully available?

Reviewer #1: Yes

Reviewer #2: Yes

5. Is the manuscript presented in an intelligible fashion and written in standard English?

Reviewer #1: Yes

Reviewer #2: Yes

6. Review Comments to the Author

Reviewer #1: Thank you for giving me the opportunity to review the manuscript again. This time, I completely agree with the content of the manuscript. The author has shown openness and acceptance of the reviewers' opinions. The edited content fully satisfies 100% of the comments. I recommend that the manuscript be considered for publication. I wish all the best for the manuscript, the author, and our journal.

Reviewer #2: (No Response)

7. PLOS authors have the option to publish the peer review history of their article (what does this mean? ). If published, this will include your full peer review and any attached files.

**Do you want your identity to be public for this peer review?** For information about this choice, including consent withdrawal, please see our Privacy Policy .

Reviewer #1: No

Reviewer #2: No

---

## [Editor Report · Acceptance letter]

PONE-D-25-15484R1

PLOS ONE

Dear Dr. Yin,

I'm pleased to inform you that your manuscript has been deemed suitable for publication in PLOS ONE. Congratulations! Your manuscript is now being handed over to our production team.

Kind regards,

on behalf of

Dr. Jasman Tuyon

Academic Editor

PLOS ONE